# Fast kinetics of magnesium monochloride cations in interlayer-expanded titanium disulfide for magnesium rechargeable batteries

Hyun Deog Yoo[1], Yanliang Liang[1], Hui Dong[1], Junhao Lin[2,3], Hua Wang[4], Yisheng Liu[5], Lu Ma[6], Tianpin Wu[6], Yifei Li[1], Qiang Ru[1], Yan Jing [1], Qinyou An[1], Wu Zhou [3], Jinghua Guo [5], Jun Lu [7], Sokrates T. Pantelides [2,3], Xiaofeng Qian[4] & Yan Yao [1,8]

Magnesium rechargeable batteries potentially offer high-energy density, safety, and low cost due to the ability to employ divalent, dendrite-free, and earth-abundant magnesium metal anode. Despite recent progress, further development remains stagnated mainly due to the sluggish scission of magnesium-chloride bond and slow diffusion of divalent magnesium cations in cathodes. Here we report a battery chemistry that utilizes magnesium mono-chloride cations in expanded titanium disulfide. Combined theoretical modeling, spectro-scopic analysis, and electrochemical study reveal fast diffusion kinetics of magnesium monochloride cations without scission of magnesium-chloride bond. The battery demon-strates the reversible intercalation of 1 and 1.7 magnesium monochloride cations per titanium at 25 and 60 °C, respectively, corresponding to up to 400 mAh g$^{-1}$ capacity based on the mass of titanium disulfide. The large capacity accompanies with excellent rate and cycling performances even at room temperature, opening up possibilities for a variety of effective intercalation hosts for multivalent-ion batteries.

[1] Department of Electrical and Computer Engineering & Materials Science and Engineering Program, University of Houston, Houston, TX 77204, USA. [2] Department of Physics and Astronomy, Vanderbilt University, Nashville, TN 37235, USA. [3] Materials Science and Technology Division, Oak Ridge National Laboratory, Oak Ridge, TN 37831, USA. [4] Department of Materials Science and Engineering, Texas A&M University, College Station, TX 77843, USA. [5] Advanced Light Source, Lawrence Berkeley National Laboratory, 1 Cyclotron Road, Berkeley, CA 94720, USA. [6] X-Ray Science Division, Argonne National Laboratory, Lemont, IL 60565, USA. [7] Chemical Sciences and Engineering Division, Argonne National Laboratory, Argonne, IL 60439, USA. [8] Texas Center for Superconductivity, University of Houston, Houston, TX 77204, USA. Correspondence and requests for materials should be addressed to Y.Y. (email: yyao4@uh.edu)

Magnesium rechargeable batteries (MRBs) are emerging as an attractive candidate for energy storage in terms of safety[1, 2], energy density[3], and scalability[4] because magnesium metal has ideal properties as a battery anode: high capacity, low redox potential, dendrite-free deposition, and earth-abundant resources. Since the first MRB prototyped by Aurbach et al.[1], significant progress has been made in cathodes[5–17], electrolytes[18–25], and anodes[26–29]. One critical challenge for MRBs is the development of Mg storage cathodes with higher capacity and operating voltage than Chevrel phase $Mo_6S_8$ cathodes[30, 31], which operate at ca. 1 V vs $Mg/Mg^{2+}$ with capacity of ca. 100 mAh $g^{-1}$. Recently, Nazar et al. reported spinel $Ti_2S_4$ and layered $TiS_2$ cathodes with a specific capacity of 200 and 160 mAh $g^{-1}$, respectively[16, 17]; however, both cathodes were operated at elevated temperature (i.e., 60 °C) due to the kinetic limitations.

Two major factors limit the development of MRB intercalation cathodes at room temperature (Fig. 1a). First, as $MgCl^+$ is the major electroactive species in typical halide-based Mg electrolytes[32–39], the Mg–Cl bond needs to be broken to free up the intercalating $Mg^{2+}$ species, which process requires a high activation energy ($E_a$) of at least 3 eV[37]. Second, most Mg-ion cathodes studied so far suffer from sluggish $Mg^{2+}$ diffusion because of the extremely high-energy barrier for $Mg^{2+}$ migration in host materials[3, 40].

In this work, we report a MRB based on a $MgCl^+$ intercalation cathode, a Mg anode, and a standard chloride-based electrolyte. Moving from the divalent $Mg^{2+}$ to the monovalent $MgCl^+$ as the charge carrier makes Mg-ions similar to one-electron-transfer alkaline metal ions where (1) only low-energy desolvation ($E_a \sim$ 0.8 eV) but not high-energy Mg–Cl scission ($E_a > 3$ eV) is necessary before intercalation and (2) the polarization strength of the ion, and hence the ion diffusion energy barrier, is low (Fig. 1b). The new battery chemistry illustrated using interlayer-expanded titanium disulfide ($TiS_2$) cathode as an example demonstrates 1 and 1.7 $MgCl^+$ intercalation per formula of $TiS_2$ at 25 and 60 °C, respectively, corresponding to high reversible capacities of up to 400 mAh $g^{-1}$ based on the mass of $TiS_2$. The electrode kinetics is fast even at room temperature. The chemical nature of intercalation species is thoroughly investigated using a combination of theoretical calculations and various spectroscopic and electrochemical studies.

## Results

**Theoretical modeling for the diffusion of $MgCl^+$ vs $Mg^{2+}$.** Although the faster diffusion owing to the lower polarity of $MgCl^+$ vs $Mg^{2+}$ is expectable based on the predictably decreased polarization strength, computational studies provide quantitative information on the diffusivity as a function of interlayer distance and the chemical structure of the ions, as well as the extent of $TiS_2$ expansion required for $Mg^{2+}$ and $MgCl^+$ to achieve maximum diffusivities. The diffusion behavior of $Mg^{2+}$ vs $MgCl^+$ in layered materials is studied using $TiS_2$ as a model compound with first-principles calculations. The diffusion of $Mg^{2+}$ in layered $TiS_2$ and its sensitivity to the interlayer spacing ($c$) have been extensively studied by previous theoretical modeling effort[41], which clearly demonstrated a significant decrease of the migration barrier with increasing lattice expansion (i.e., from 5.7 to 6.3 Å). Herein, we study the effect of further expansion of $TiS_2$ on the mobility of $Mg^{2+}$ and $MgCl^+$. As $c$ increases from 5.7 to 10.9 Å, the $Mg^{2+}$ migration barrier reduces from 1.06 to 0.51 eV (Fig. 2a, b) as a result of smaller total binding energy between Mg and S in $TiS_2$, in excellent agreement with the previous work[41]. However, further expansion from 10.9 Å could not reduce the barrier anymore (Fig. 2a and Supplementary Fig. 1). In contrast, when $MgCl^+$ is considered as the active diffusive species in the interlayer-expanded $TiS_2$ ($c = 10.9$ Å), migration barrier could be drastically reduced to 0.18 eV. Assuming the standard Arrhenius expression ($D \propto e^{-E_a/kT}$), the barrier decrease from 0.51 to 0.18 eV is equivalent to $4 \times 10^5$ times faster cation diffusion at room temperature.

To understand the striking difference between the migration barrier for $Mg^{2+}$ and $MgCl^+$, we studied the electron density difference for the above three scenarios (Fig. 2c): (a) Mg@$TiS_2$ with $c = 5.7$ Å, (b) Mg@$TiS_2$ with $c = 10.9$ Å, and (c) MgCl@$TiS_2$ with $c = 10.9$ Å. In the case of (a), with Mg initially bonded with both top and bottom $TiS_2$ layers, electrons of $TiS_2$ were largely polarized toward Mg with notable amount accumulated around the six neighboring S atoms. In the case of (b), the coordination number of Mg reduces from six to three, causing a net reduction in the electrostatic interaction. Although the electron density around the three neighboring S atoms increases (manifested by the slightly enlarged orange lobes), the net migration energy barrier is reduced because of the decrease in the coordination

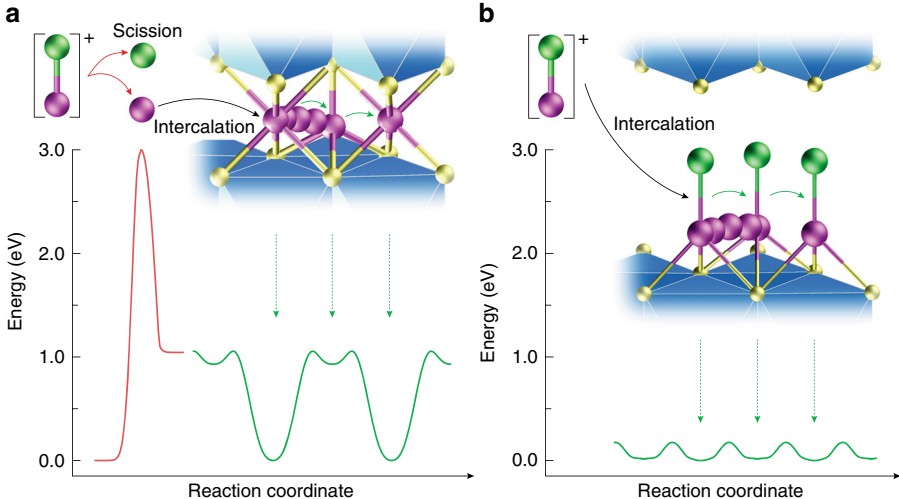

**Fig. 1** Energy diagrams for the intercalation and diffusion of $Mg^{2+}$ and $MgCl^+$. **a** Typical intercalation of $Mg^{2+}$ involves scission of $MgCl^+$ ions into $Mg^{2+}$ and $Cl^-$, which requires substantial activation energy of 3 eV at least. Subsequent diffusion of divalent $Mg^{2+}$ also has a high-migration energy barrier of 1.06 eV, which results in the limited level of intercalation at room temperature. **b** Intercalation of $MgCl^+$ bypasses the sluggish scission of the Mg–Cl bond at the electrolyte–cathode interface; afterwards $MgCl^+$ diffuses fast in the expanded interlayers due to the fairly low-migration energy barrier of 0.18 eV. Mg and Cl atoms are shown as *purple* and *green spheres*, respectively

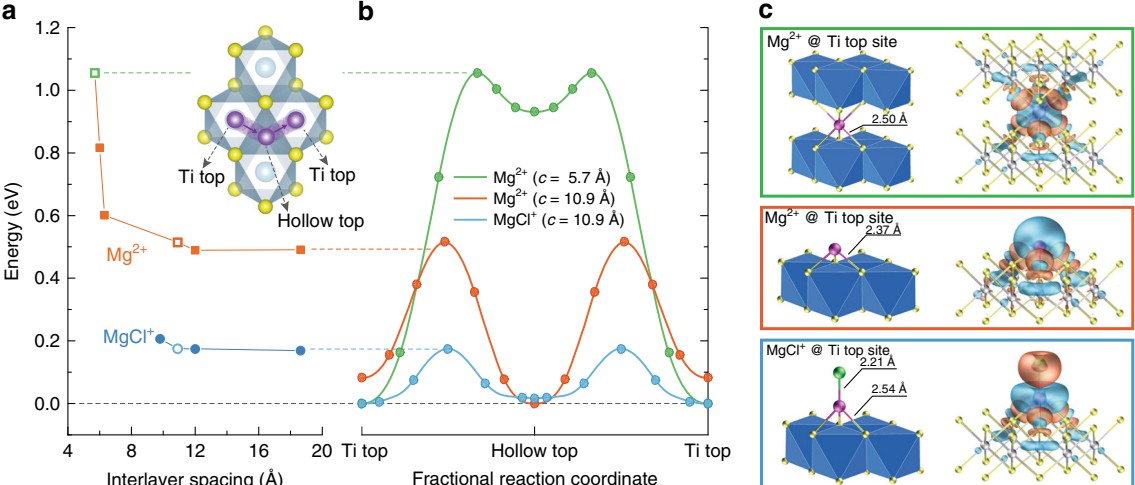

**Fig. 2** First-principles calculations for the diffusion of Mg-ions in $TiS_2$. **a** Energy barrier for the migration of $Mg^{2+}$ and $MgCl^+$ as a function of the interlayer distance of $TiS_2$ at the dilute limit. The diffusion path from a Ti top to another Ti top site via the adjacent Hollow top site is shown in the inset. **b** Energy diagrams along the diffusion path for the three representative cases of $Mg^{2+}$ at $c = 5.7$ Å (*green*), $Mg^{2+}$ at $c = 10.9$ Å (*orange*), and $MgCl^+$ at $c = 10.9$ Å (*cyan*). **c** Atomic configurations of $Mg^{2+}$ and $MgCl^+$ at Ti top site for the three cases in **b**. The *right panels* show the charge difference plots constructed by subtracting the valence electron density of individual Mg atom, Cl atom, and $TiS_2$ layers from that of Mg@$TiS_2$ or MgCl@$TiS_2$. *Blue* and *orange colors* represent depletion and accumulation of electron, respectively

number. In the case of (c), the bonding of the negatively charged Cl to Mg makes the electrons of $TiS_2$ less polarized toward Mg, as is indicated by shrunk orange lobes around the three coordinating S atoms. As a result, the strength of the three Mg–S bonds reduces, as is indicated by the increase in the Mg–S bond length from 2.37 Å to 2.54 Å. The reduced strength of Mg–S bonds leads to much smaller migration energy barrier. These results demonstrate that the marked reduction of migration barrier comes from two equally important factors: the increase of interlayer distance and the chemical structure of diffusive species involved in the migration process. It is worth noting that, without the expansion of $TiS_2$, neither the intercalation nor the fast diffusion of $MgCl^+$ would be possible. This simulation motivates us to find a method to expand the interlayer spacing to accommodate MgCl-ions.

**In situ expansion of $TiS_2$ in Mg battery cells**. In most cases, layered materials are expanded ex situ[42–44], i.e., by mechanical or chemical processes before they are introduced in a battery. However, moisture-sensitive $TiS_2$ is prone to oxidation during the ex situ processes and subsequent cell fabrication. It has been reported that intercalation of organic compounds can expand layered materials in a highly controllable manner without exfoliating the structure into single layers[45–47]. We chose the chemically stable 1-butyl-1-methylpyrrolidinium ion ($PY14^+$) as an organic "pillar"[48, 49], which expands $TiS_2$ layers in situ, i.e., by discharging a complete $TiS_2$/Mg cell using the electrolyte containing $PY14^+$ ions. The reversible Mg deposition and dissolution in the electrolyte solution is not hampered with the addition of the $PY14^+$ ions (Supplementary Table 1 and Supplementary Fig. 2).

In operando X-ray diffraction (XRD) shed light on the structural evolution of $TiS_2$ during the initial activation of the cell (Fig. 3a and Supplementary Fig. 3). The as-fabricated electrode (stage 0) shows a peak at 15.56°, corresponding to the (001) plane of pristine $TiS_2$ with $c = 5.69$ Å. After discharging to 1 V vs Mg/$Mg^{2+}$ (stage 1), new peaks evolve at 8.13° and 16.31°, corresponding to (001) and (002) planes with $c = 10.87$ Å. Further discharging to 0.2 V vs Mg/$Mg^{2+}$ (stage 2) results in four new peaks at 4.74°, 9.49°, 14.26°, and 19.04°, corresponding to (001) to

(004) planes with $c = 18.63$ Å (Supplementary Table 2). The shifts of diffraction peaks from stage 0 to 1 and from stage 1 to 2 are irreversible (Supplementary Fig. 4). And the expanded interlayer spacing is maintained same as that of stage 2 upon subsequent stages of discharge/charge cycling. Deeper discharging to 0 V vs Mg/$Mg^{2+}$ (stage 3) does not further shift the peaks but the peak intensities become attenuated, suggesting a structural disorder as evidenced by intralayer ruptures in the scanning transmission electron microscopy (STEM) image at stage 3 (Fig. 3b). The interlayer spacing from the STEM image for each stage is in excellent agreement with the value from XRD.

High-energy X-ray diffraction (HE-XRD) confirms the interlayer distance in Supplementary Table 2 with higher resolution (Fig. 3c). Moreover, the HE-XRD patterns at four stages show that (100) and (110) peaks, solely related with *ab*-plane, do not shift, indicating the intralayer structure of $TiS_2$ is preserved during expansion along the *c*-direction. Cross-sectional elemental mapping at stage 4 shows alternating layers of Ti and C, which is a clear evidence that organic $PY14^+$ "pillars" stay in the van der Waals gap of $TiS_2$ after discharge (Fig. 3d). The expanded $TiS_2$, or ex$TiS_2$, remains compact without exfoliation during the cycling (Supplementary Fig. 5). The initial activation is complete at this stage with $PY14^+$ contributing a one-time irreversible capacity of ~50 mAh $g^{-1}$. And $PY14^+$ does not contribute to the reversible capacity in the following cycles.

**Chemical nature of ex$TiS_2$ at each stage of intercalation**. We conducted detailed characterizations of samples prepared at different stages. First, intercalating species at each stage was investigated using energy dispersive spectroscopy (EDS), inductively coupled plasma optical emission spectrometry (ICP-OES), and X-ray photoelectron spectroscopy (XPS). Although negligible Mg and Cl signals were detected at stage 1 (Fig. 4a, b), strong N 1s peak in the XPS spectrum confirms that only $PY14^+$ ions intercalate during stage 0 to 1 (Fig. 4a). Signals for Mg and Cl atoms increase substantially when further discharging to stages 2 and 3; and both signals decrease when charging back to stage 4 (Fig. 4a, b and Supplementary Table 3). The atomic ratio of Mg to Ti reaches $1 \pm 0.1$ at stage 3 according to ICP-OES, whereas the atomic ratio of Mg to Cl is $1 \pm 0.2$ at each stage according to EDS

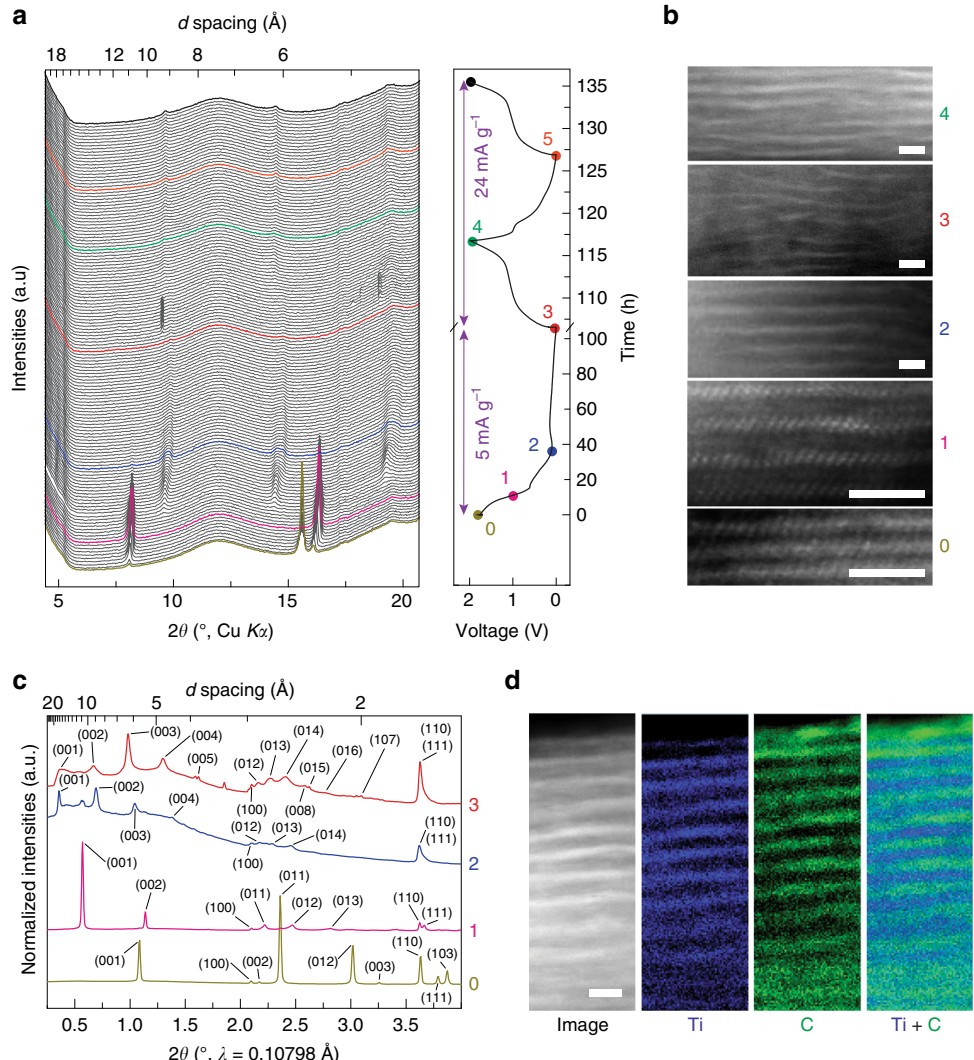

**Fig. 3** Structural characterizations of $TiS_2$ during the initial activation. **a** In operando XRD characterization and corresponding galvanostatic voltage profile for the first two cycles. **b** STEM images for stages 0 to 4. **c** HE-XRD patterns for stages 0 to 3. **d** The STEM image and the elemental mapping of Ti and C at stage 4. Scale bars: 2 nm

(Supplementary Table 3). Electron energy loss spectroscopy (EELS) also evidences Mg and Cl $L_{2,3}$ peaks for stage 3, whereas the peaks are absent at stage 4 (Fig. 4c). Both discharged and charged electrodes (stages 3 and 4, respectively) contain ~1 at.% of Al, which most likely originated from $AlPh_2Cl_2^-$ anions adsorbed on the surface (Fig. 4b). Proton nuclear magnetic resonance ($^1$H-NMR) spectroscopy detected small amount of tetrahydrofuran (THF) from stages 2 to 4, suggesting possible solvent co-intercalation in exTiS$_2$ (Supplementary Fig. 6). The number of THF molecule in the fully discharged sample (~0.16 per $MgCl^+$) is far lower than what is typically needed for the solvation of $MgCl^+$ ions (e.g., 3 in $[MgCl\cdot3THF]^+$). Combining these results and thermogravimetric analysis (Supplementary Fig. 7 and Supplementary Table 4), we obtain the composition of the discharged compound at stage 3 as $(MgCl)_{1.0}TiS_2[(PY14)_{0.20}(THF)_{0.16}]$.

Second, near-edge X-ray absorption fine structure (NEXAFS) of Mg $K$-edge reveals the coordination state of the inserted MgCl-ions (Fig. 4d). Recent experimental and theoretical works showed that at least 1 eV lower onset of X-ray absorption for tetracoordinated Mg-ions (e.g., $[Mg_2Cl_2\cdot4THF]^{2+}$ or $[MgCl\cdot3THF]^+$) compared with the hexacoordinated Mg-ions (e.g., $[Mg_2Cl_3\cdot6THF]^+$)[33, 50, 51]. The onset energy for the

magnesiated $TiS_2$ at stage 3 is closest to that of the tetra-coordinated $[Mg_2Cl_2\cdot4THF]^{2+}$[50]. Therefore, the intercalated MgCl-ions maintain tetracoordination of Mg with 1 Cl and 3 S atoms as predicted in Fig. 2c.

Last, to probe the sulfur coordination environment change upon $MgCl^+$ intercalation, S $K$-edge NEXAFS was performed on $(MgCl)_xTiS_2$ for $x = 0$, 0.5, and 1. The NEXAFS spectra are displayed in Fig. 4e after background subtraction and normalization. Three S $K$-edge peaks $C$, $C'$, and $D$ represent the transitions from S $1s$ to S $3p$ orbitals. The $C$ and $C'$ peaks can be assigned to $t_{2g}$ and $e_g$ states from the hybridization of S $3p$ and Ti $3d$ orbitals via $\pi^*$ and $\sigma^*$ antibonding, respectively. The intensity and width of $C$ and $C'$ peaks decrease upon $MgCl^+$ intercalation, whereas peak $C$ exhibits more pronounced decrease in the intensity than $C'$; but no noticeable energy shift is observed for the two peaks. Peak $D$, which can be assigned to hybridized S $3p$ and Ti $4s$ and $4p$ orbitals, shows a progressive shift toward lower energy and an increase in the intensity upon $MgCl^+$ intercalation. The observed spectral changes are similar to the experimental and theoretical study for $Li^+$ intercalation into $TiS_2$[52], in which the reduced intensity of $C$ and $C'$ peaks was originated from the structural distortion, with more pronounced influence on peak $C$ due to the partial filling of the $t_{2g}$ states by the charge transfer

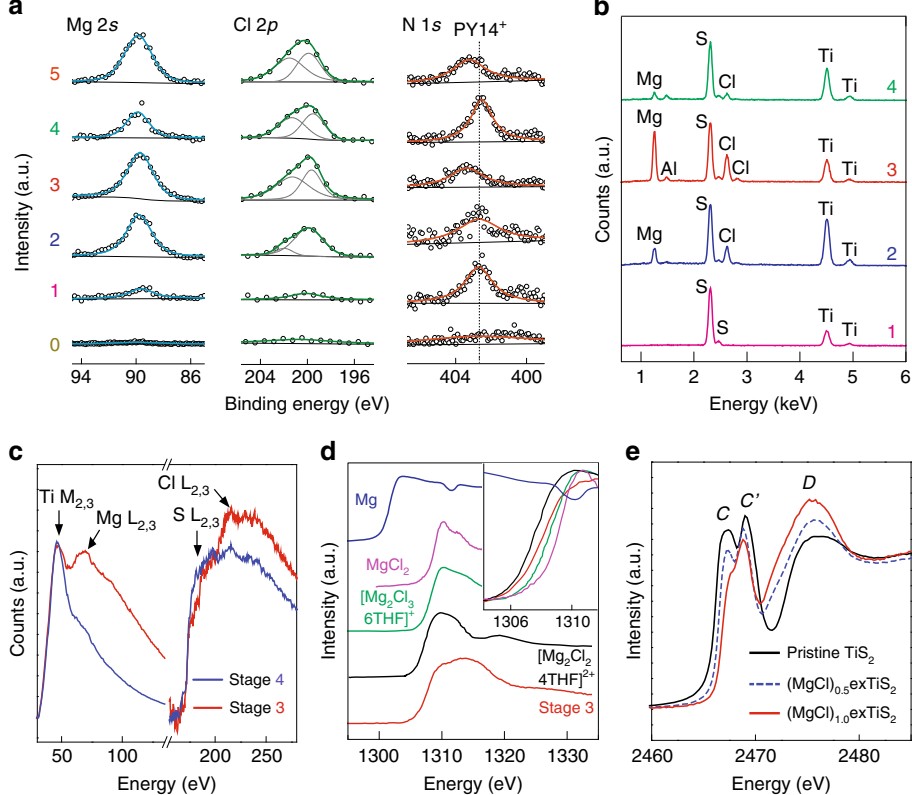

**Fig. 4** Chemical nature of the intercalation compound at each stage. **a** XPS spectra of Mg 2*s*, Cl 2*p*, and N 1*s* for stages 0 to 5. **b** EDS spectra for stages 1 to 4. **c** EELS spectra at stages 3 and 4. **d** Mg *K*-edge NEXAFS spectra of Mg metal, $MgCl_2$ powder, $[Mg_2Cl_2 \cdot 4THF]^{2+}$ (Nakayama et al.[50]), $[Mg_2Cl_3 \cdot 6THF]^+$ (Benmayza et al.[51]), and magnesiated $exTiS_2$ at stage 3. **e** Experimental S *K*-edge NEXAFS spectra for $TiS_2$ (*black*), $(MgCl)_{0.5}exTiS_2$ (*blue, dashed*), and $(MgCl)_{1.0}exTiS_2$ (*red*)

upon intercalation of ions. In our case, the structural distortion and charge transfer would be results from the $MgCl^+$ intercalation. The change of peak *D* may reflect the bonding between Mg and S atoms and the coordination number of S changes from three in $TiS_2$ to six in $(MgCl)_x exTiS_2$, whereby the hybridization of S increases.

**Electrochemical performances of exTiS₂/Mg battery cells.** $exTiS_2$ shows a highly reversible capacity of 239 mAh g$^{-1}$ based on the mass of $TiS_2$ at the current density of 24 mA $g_{TiS_2}^{-1}$ (0.1C-rate), or 173 mAh g$^{-1}$ based on the composite mass of $TiS_2[(PY14)_{0.20}(THF)_{0.16}]$ at room temperature (Fig. 5a). Overall, 70% of the capacity can be maintained (179 mAh $g_{TiS_2}^{-1}$) at a much higher current of 240 mA g$^{-1}$ (1C). The capacity of 239 mAh g$^{-1}$ corresponds to 1 $MgCl^+$ per formula of $TiS_2$, which is in accordance with the atomic ratio of 1 for Mg/Ti and Mg/Cl from ICP-OES and EDS measurements, respectively. The capacity values are higher than that of state-of-the-art cathodes operated at room temperature: e.g., Chevrel phase $Mo_6S_8$ (95 mAh g$^{-1}$ at 0.1C)[53], thiospinel $Ti_2S_4$ (130 mAh g$^{-1}$ at 0.02C)[16], and layered $TiSe_2$ (110 mAh g$^{-1}$ at 0.05C)[11]. The volumetric capacity of $exTiS_2$ is 235 Ah L$^{-1}$, which is 3.6 times as high as that of pristine $TiS_2$ (66 Ah L$^{-1}$) but 55% lower than that of $Mo_6S_8$ (519 Ah L$^{-1}$) due to the decreased density caused by the volume expansion (Supplementary Table 5). The sloping shape of the discharging voltage profiles suggest formation of $(MgCl)_x exTiS_2$ solid solution. This sloping shape agrees with the theoretical calculation for $Mg^{2+}$ intercalation into layered $TiS_2$[41], but the voltage is lower than the calculated value probably because the interlayer expansion weakens the intercalation energy[54]. In terms of cycling

stability, the $exTiS_2$ electrode exhibits 80% capacity retention after 400 cycles at 1C-rate with coulombic efficiency consistently higher than 99% (Fig. 5b).

To confirm the mechanism is indeed intercalation rather than surface adsorption, cyclic voltammetry (CV) was measured at scan rates (*v*) from 0.1 to 10 mV s$^{-1}$ (Supplementary Fig. 8). Figure 5c shows the linear relationship between peak current vs $v^{1/2}$, indicating the mechanism is indeed diffusion-limited intercalation rather than surface-limited adsorption. Electrochemical impedance spectroscopy (EIS) was measured to check the capacitance at stages 0–3. The significant increase in capacitance at stage 3 can be interpreted as the larger interfacial area compared to stages 0–2, which may be related to the intralayer ruptures at stage 3 (Supplementary Fig. 9). However, the capacitance of 60 F g$^{-1}$ corresponds to the capacity of ca. 33 mAh g$^{-1}$, which is only 14% of the total capacity. Such observation also supports the conclusion that most of the capacity comes from intercalation rather than adsorption.

To exclude the effect of $PY14^+$ ions in the electrolyte on electrochemical performance of $exTiS_2$, the electrode at stage 4 (i.e., completely deintercalated one) was transferred into a new cell with standard APC electrolyte solution without $PY14^+$ (Supplementary Fig. 10). The performance is largely retained with the reversible capacity of about 200 mAh g$^{-1}$, which is 10 times larger than the capacity before the expansion. The decrease in capacity compared with the original cell is most likely due to the inevitable material loss during the thorough washing step before transferring to the new cell. This result confirms that the electrochemical performances of $exTiS_2$ come from (de)intercalation of $MgCl^+$ and are independent of $PY14^+$ ions in the electrolyte. Supplementary Fig. 11 shows stable cycling of 80%

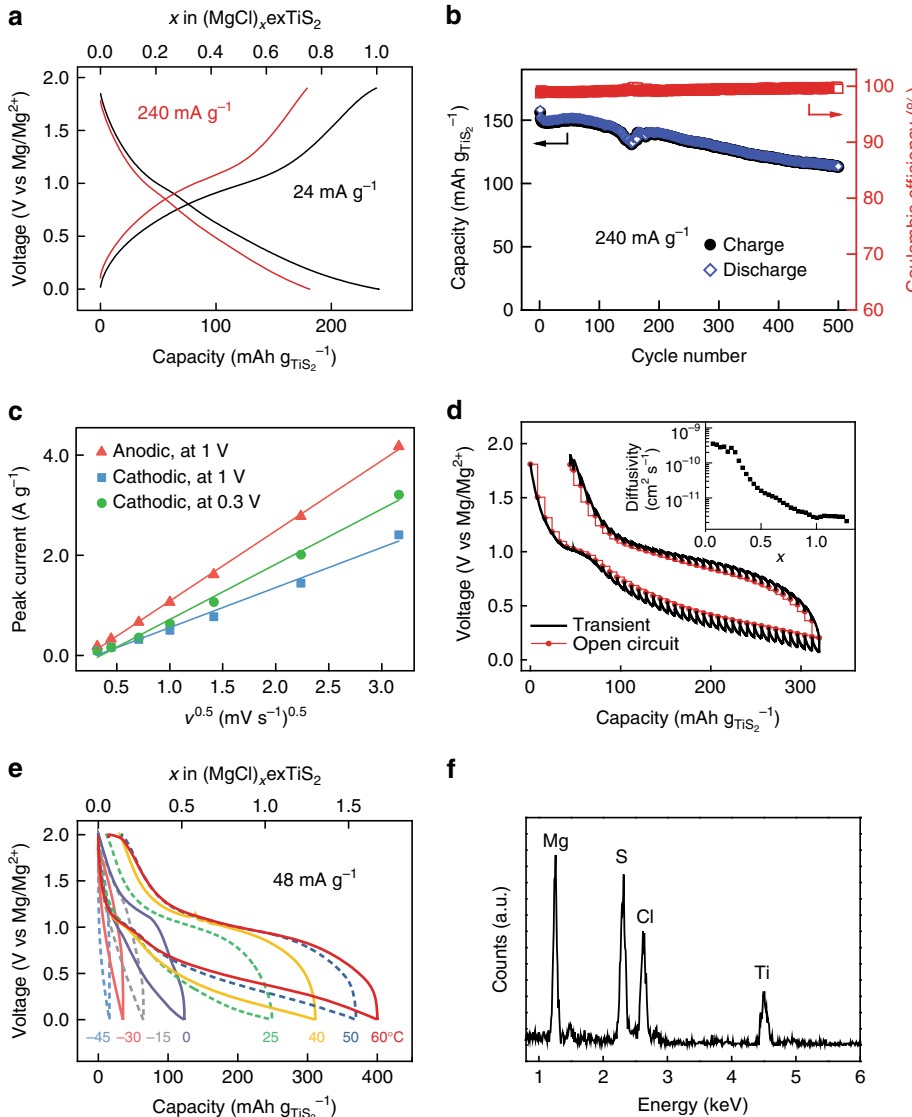

**Fig. 5** Electrochemical performances of exTiS$_2$. **a** Galvanostatic voltage profiles of the exTiS$_2$ electrode at 24 and 240 mA g$^{-1}$ at 25 °C. The number of MgCl$^+$ intercalation per exTiS$_2$ is also shown in the top axis. **b** Cycling performance at 1C-rate (i.e., 240 mA g$^{-1}$). The capacity dip at the 150th cycle is due to temperature change caused by temporary failure of air conditioner. **c** A linear relationship between the peak current in the cyclic voltammogram and the square root of the scan rate (*v*). **d** GITT curve of an exTiS$_2$ electrode. **e** Voltage profiles of exTiS$_2$ electrodes at temperatures varied from −45 to 60 °C at 48 mA g$^{-1}$. **f** EDS spectra for exTiS$_2$ discharged at 60 °C. Specific capacity is calculated based on the mass of TiS$_2$

capacity retention after 350 cycles, similar to the cyclability observed in the PY14$^+$-containing electrolyte. This result reaffirms that the organic cations are chemically stable and sufficiently immobile and stay in the structure with no change during the cycling.

Galvanostatic intermittent titration technique (GITT) was used to probe the reaction mechanism and determine the diffusivity of MgCl$^+$ in exTiS$_2$ as a function of depth-of-discharge (Fig. 5d)[55]. The open circuit potential (denoted as *red*) during the cycling corresponds to the true thermodynamic voltage profile. There is a voltage gap between charge and discharge, reflecting a MgCl$^+$ (de) intercalation mechanism that involves the redistribution of a second mobile yet much more sluggish species, e.g., PY14$^+$, in the interlayer[56]. The diffusivity calculated during discharge is initially high at the level of $3 \times 10^{-10}$ cm$^2$ s$^{-1}$ but decreases with increasing MgCl$^+$ concentration and then stays constant as $3 \times 10^{-12}$ cm$^2$ s$^{-1}$ towards the end of discharging process (*inset* of Fig. 5d). The decrease in the diffusivity with increasing MgCl$^+$ concentration

can be due to a divacancy diffusion mechanism as is well known to occur in Li$^+$ intercalation compounds[57]. The average MgCl$^+$ diffusivity of $10^{-11}$ cm$^2$ s$^{-1}$ is one order of magnitude higher than that of Mg$^{2+}$ in poly(ethylene oxide)-intercalated MoS$_2$[58] and Chevrel phase Mo$_6$S$_8$[59]. The fast kinetics of MgCl$^+$ diffusion in exTiS$_2$ agrees with the simulation results. The high diffusivity of MgCl$^+$ ions in exTiS$_2$ interlayers enables the larger specific capacity and higher rate capability compared to other Mg$^{2+}$ intercalation hosts reported at room temperature.

Finally, the effects of temperature on exTiS$_2$/Mg cells were studied. As the temperature increased from −45 to 60 °C, MgCl$^+$ intercalation capacity increases significantly (Fig. 5e). This improvement can be attributed to increased MgCl$^+$ diffusivity; considering the migration barrier of 0.18 eV for MgCl$^+$ in exTiS$_2$, the diffusivity increases to 209% when temperature increases from 25 to 60 °C according to Arrhenius relation with temperature ($D \propto e^{-E_a/kT}$). At 60 °C, the cell reaches a capacity of 400 mAh g$_{TiS_2}^{-1}$ (394 Ah L$^{-1}$), or 269 mAh g$^{-1}$ based on the

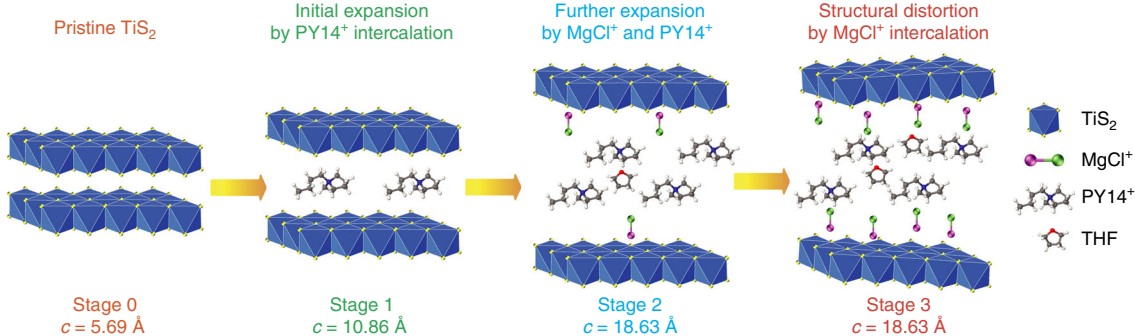

**Fig. 6** A schematic of structural evolution of $TiS_2$ at different stages of intercalation. Interlayers are expanded or distorted as different amount of pillaring molecules, complex cations, and solvents are intercalated into the van der Waals gap of a host material at each stage

mass of the composite, corresponding to the intercalation of 1.7 $MgCl^+$ per unit $TiS_2$. Discharging at a lower rate (24 mA g$^{-1}$) at 60 °C yielded the capacity of 450 mAh g$^{-1}$ (1.9 $MgCl^+$ per $TiS_2$), which was in accordance with Mg/Ti ratio of 2.1 ± 0.3 from the EDS spectrum (Fig. 5f). Having more than one electron reversibly stored per $TiS_2$ formula is to our knowledge unprecedented. Each 1T-$TiS_2$ unit possesses two distinguishable sites: the octahedral (Ti top) and tetrahedral (hollow top) sites. Without interlayer expansion, only one site is energetically viable for ion intercalation (octahedral site in this case). Intercalation of a monovalent ion in a tetrahedral site only happens at an exceedingly low potential[60] (−0.27 V vs Mg/Mg$^{2+}$) and the process is not reversible (Supplementary Fig. 12). When $TiS_2$ is expanded, intercalation at both sites will exhibit similar energy levels per our simulation results (Fig. 2b), thereby enabling the intercalation of more than one $MgCl^+$ per unit $TiS_2$. In summary, the $MgCl^+$ intercalation into ex$TiS_2$ is fully reversible over a wide temperature range from −45 to 60 °C. Higher temperature leads to higher $MgCl^+$ diffusivity and nearly doubled reversible capacity.

## Discussion

A schematic illustration in Fig. 6 summarizes the four stages of intercalation. From stage 0 to 1, the interlayer distance increases from 5.69 Å to 10.86 Å due to the intercalation of PY14$^+$ ions, which act as pillars to expand $TiS_2$ and make it possible for $MgCl^+$ ions to intercalate inside the gap. From stage 1 to 2, the interlayer distance further increases to 18.63 Å as $MgCl^+$ ions begin to intercalate thanks to the initial expansion by PY14$^+$ ions. A small number of THF molecules are intercalated during this step. From stage 2 to 3, a large number of $MgCl^+$ ions are intercalated and the corresponding mechanical stress results in structural distortions (Fig. 3b). Following these steps, PY14$^+$ ions stay inside the van der Waals gap, whereas the highly mobile $MgCl^+$ is (de)intercalated reversibly. After the completion of the first discharge process, the interlayer-expanded $TiS_2$ cathode shows a reversible capacity as high as 239 mAh g$^{-1}$ (Fig. 5a). Completing stage 3 was essential in realizing such a large reversible capacity; stopping at stage 2 led to a lower reversible capacity of ca. 60 mAh g$^{-1}$ even though $TiS_2$ is expanded to the same interlayer distance for stages 2 and 3 (Supplementary Fig. 13). It is noteworthy that the theoretical predictions suggest no difference in the diffusivity of $MgCl^+$ as long as the interlayer distance is larger than 10.9 Å. However, the theoretical modeling does not account for the steric hindrance of the intercalated PY14$^+$ cations. Therefore, in reality, fast diffusion of $MgCl^+$ may require more substantial structural adjustments (e.g., interlayer expansion to 18.6 Å, intralayer ruptures as shown in Fig. 3b, etc.), which make the structure more accessible to $MgCl^+$ ions. Further engineering of less bulky and more light-weight pillar species may require

smaller structural adjustments and also lead to higher specific capacity (which is determined by the total mass of the host and the pillar).

The following electrochemical reactions are proposed for room temperature:

$$Cathode: exTiS_2 + MgCl^+ + e^- \leftrightarrow (MgCl)exTiS_2, \quad (1)$$

$$Anode: nMg + Mg_xCl_y^{z+} \leftrightarrow yMgCl^+ + 2ne^-, \quad (2)$$

$$Overall: 2exTiS_2 + Mg + 1/nMg_xCl_y^{z+} \leftrightarrow z/nMgCl^+ + 2(MgCl)exTiS_2 \quad (3)$$

where $Mg_xCl_y^{z+}$ refers to $MgCl_2$, $Mg_2Cl_3^+$, $Mg_3Cl_5^+$, etc. so that $x$, $y$, $z$, and $n$ are whole numbers that satisfy $y > x$, $z = 2x−y = 0$ or 1, and $n = y−x = 1$, 2, 3, etc. Equation (1) describes the reversible intercalation of $MgCl^+$ into ex$TiS_2$ at the cathode side. This complex-ion intercalation mechanism is similar to the $AlCl_4^-$ intercalation in graphite reported by Dai et al.[61], except that the intercalation of $MgCl^+$ happens during the reduction of the host rather than the oxidation. Equation (2) describes the simultaneous generation of $MgCl^+$ at the Mg anode by converting $Mg_xCl_y^{z+}$ species in the electrolyte. The overall battery reaction in Eq. (3) can continue as long as $Mg_xCl_y^{z+}$ species are available in the electrolyte. Equation (3) depicts that $MgCl_2$ is converted to intercalated $MgCl^+$ without leaving anything behind because it is neutral ($z = 0$), whereas complex cations with $y > x$ are converted leaving $MgCl^+$ of same $z/n$ equivalents in the electrolyte. Therefore, the cation concentration of the electrolyte stays unchanged throughout the reaction in either case.

According to Eq. (3), two moles of (MgCl)$TiS_2$ are formed with the consumption of one mole of magnesium-chloride ($MgCl_2$). Considering a 1 M concentration for $MgCl_2$[62], the necessary volume of electrolyte to match an areal capacity of 1 mAh cm$^{-2}$ (which approximates the areal capacity of a practical cell) is 18.7 μL cm$^{-2}$. This amount of electrolyte is sufficiently small to be accommodated in practical batteries; commercial separators with 300 μm thickness and 80% porosity may uptake 24 μL cm$^{-2}$ of electrolyte[63]. Further scale-up of the battery cell can be supported by adding $MgCl_2$, which is the main neutral species in the chloride-based electrolytes[20, 64, 65]. Dissolved $MgCl_2$ can participate in the battery reaction of Eq. (2) directly, or indirectly by buffering $Mg_2Cl_3^+$ through the dynamic equilibrium among $MgCl_2$, $MgCl^+$, and $Mg_2Cl_3^+$ species[36]. Recent reports show the solubility of $MgCl_2$ can be greatly increased with the aid of diorganomagnesium compounds[62].

The MgCl-ion storage mechanism can be generalized to other two-dimensional materials. For example, molybdenum disulfide ($MoS_2$) also demonstrated ca. 280 mAh g$^{-1}$ after 10 cycles in the

APC electrolyte containing PY14$^+$ ions (Supplementary Fig. 14). The average discharge voltage of 0.7 V in this work needs to be increased in future studies by exploring higher voltage cathodes. Attempts to use the same approach for high-voltage cathode (e.g., layered vanadium oxide) was not quite successful so far due to the limitation of the nucleophilic nature of the APC electrolyte, which reacts chemically with oxides. It is worthwhile to re-examine this method to layered oxide cathodes, when non-nucleophilic electrolytes with higher voltage stability window become widely available[24].

Recent years have seen increasing concerns about potential corrosion problems related to chloride-containing electrolytes. Oh and colleagues[66] have reported that PY14Cl is an effective inhibitor against the corrosion by Cl$^-$ at high potentials. Therefore, PY14$^+$ is not only a pillar for expanding TiS$_2$ but also a corrosion inhibitive additive. Meanwhile, halogen-free electrolytes are under development for even wider voltage windows[24]. In those cases, the anion (A$^-$) can associate with Mg$^{2+}$ to form MgA$^+$ ions, which can be used as the charge carrier in expanded materials. In this sense, the present study provides general guidelines and design principles for the intercalation of such generalized Mg complex ions into expanded interlayers.

In summary, we report a magnesium battery chemistry enabled by MgCl$^+$ intercalation mechanism. A class of two-dimensional host materials with electrochemically expanded interlayer spacing allow intercalation of the large MgCl$^+$. With the expanded cathode, the reversible capacity and rate performance of an exTiS$_2$/Mg full cell surpass those of state-of-the-art MRBs. This work unravels the factors that determine the diffusion of ions in layered materials with respect to the interlayer distance and chemical interactions. And a new direction is identified toward overcoming the challenge of high-migration energy barrier and kinetically sluggish dissociation processes in MRBs. This battery chemistry can be extended to the intercalation of a wide range of multivalent ions (e.g., Zn$^{2+}$, Ca$^{2+}$, Al$^{3+}$) into various two-dimensional materials, highlighting the importance of an unexploited route of materials design for multivalent-ion batteries.

## Methods

**First-principles calculations**. Minimum energy pathway and the corresponding migration energy barrier were calculated using the climbing image nudged elastic band method[67]. The total energy calculations of each image along the pathway were performed in a supercell geometry described below using first-principles density functional theory as implemented in the Vienna Ab-initio Simulation Package (VASP)[68]. More specifically, we adopted the projector-augmented wave method[69], a plane wave basis with kinetic energy cut-off of 400 eV, a Γ-centered Monkhorst–Pack $k$-point sampling of $3 \times 3 \times 1$ for Brillouin zone integration[70] in the supercell calculations, and the exchange-correlation functional in the Perdew–Berke–Ernzerhof (PBE)[71] form within the generalized gradient approximation (GGA)[72]. In addition, van der Waals interaction was approximated by including the optB88-vdW nonlocal correlation functional[73]. The calculated $a$ and $c$ lattice parameter of bulk 1T-TiS$_2$ are 3.414 Å and 5.273 Å. A double layer supercell consisting of $4 \times 4 \times 2$ TiS$_2$ unit cells with one Mg$^{2+}$ was constructed to determine the migration barrier in the dilute limit with varied $c$ lattice constant of 5.7–6.3 Å. For larger $c$ lattice constant of 9.8–18.6 Å, a single layer supercell of $4 \times 4 \times 1$ TiS$_2$ unit cells with one Mg$^{2+}$ or MgCl$^+$ was constructed. At such large $c$ lattice constant, calculation with either single or double layer supercell resulted in almost same migration barrier. The charge difference plots were calculated by subtracting the valence electron density of isolated neutral Mg and Cl atoms and pristine TiS$_2$ layers from that of Mg@TiS$_2$ and MgCl@TiS$_2$ using an isosurface value of 0.015 $e$ Å$^{-3}$.

**Materials preparation**. Layered TiS$_2$ (99.8%, Strem Chemical Inc.) with an average particle size of 10 μm was used as purchased. A slurry of active material (70 wt.%), Super-P carbon (20 wt.%), and polyvinylidene fluoride (10 wt.%) dispersed in $N$-methyl-2-pyrrolidone was spread on a piece of stainless steel mesh (400 mesh, 0.8 cm$^2$) and dried as the working electrode with active material mass loading of 0.5–1 mg cm$^{-2}$. To prepare samples for analysis, we prepared electrode by cold pressing 7 mg of TiS$_2$ powders onto stainless steel mesh at 10 MPa without using binder or conductive agent. Freshly polished magnesium foil (50 μm thick, 99.95%,

GalliumSource, LLC) was used as both the counter and reference electrodes in 2- or 3-electrode cell tests. Standard APC electrolyte, a solution of 0.25 M [Mg$_2$Cl$_3$]$^+$[AlPh$_2$Cl$_2$]$^-$ in THF, was prepared following Aurbach et al. and was used as the Mg-ion electrolyte throughout this work[19]. PY14$^+$ ion was added in the APC electrolyte by dissolving 1-butyl-1-methylpyrrolidinium chloride (PY14Cl, >98%, TCI America Co.) to the concentration of 0.2 M.

**Electrochemical tests**. 2- and 3-electrode coin cells were fabricated in an Ar-filled glove box using a magnesium foil as the anode, a glass fiber separator (210 μm thick, grade 691, VWR Co.) and a tri-layer polypropylene/polyethylene/polypropylene (25 μm thick, Celgard 2325, Celgard LLC.) as the separators, and TiS$_2$ or exTiS$_2$ as the cathode. The electrochemical measurements for CV, EIS, and GITT were carried out in a specially designed 3-electrode coin cell (Supplementary Fig. 15) to measure the potential of cathode vs Mg/Mg$^{2+}$. For the 3-electrode configuration, a ring-shaped magnesium foil was used as the reference electrode connected out of the coin cell by polypropylene coated stainless steel foil. The electrochemical characterizations were conducted using a potentiostat (VMP-3, Bio-Logic Co.) and battery cyclers (CT2001A, Lanhe Co.) at room temperature. EIS was measured at a fixed potential of 1.8 V vs Mg/Mg$^{2+}$ by applying small sinusoidal potential with amplitude of 7 mV and frequency ($f$) ranging from 200 kHz to 2 mHz. Capacitance was obtained as a function of frequency by calculating the real part of the complex capacitance[74]. Before the electrochemical cycling, all the cells were activated at 25 °C by discharging the TiS$_2$/Mg cells to 0 V vs Mg/Mg$^{2+}$ at 5 mA g$^{-1}$ for ca. 100 h and subsequent cycling the cells within 0–2 V vs Mg/Mg$^{2+}$ at 24 mA g$^{-1}$ for 10 cycles (as shown in Fig. 3a). Then the activated exTiS$_2$/Mg cells were cycled within 0–2 V vs Mg/Mg$^{2+}$ at varied temperature from –45 to 60 °C. The capacity was calculated based on the mass of TiS$_2$ in the electrode unless otherwise specified; the specific capacity of 400 mAh g$^{-1}$ translated to 1.7 MgCl$^+$ per TiS$_2$ by comparing with the theoretical value of 239.3 mAh g$^{-1}$ for the intercalation of 1 MgCl$^+$ into TiS$_2$. GITT was performed by applying constant current of 24 mA g$^{-1}$ for 20 min followed by 30 min of open circuit period upon charging and discharging.

**Materials characterization**. The electrodes were characterized by scanning electron microscopy (SEM, LEO Gemini 1525, ZWL Co.), EDS (Oxford Instruments Co.), XPS (Physical Electronics Model 5700), ICP-OES (Agilent Technologies, Model 725), NMR (Oxford Instruments EUR0059), and STEM (Nion Co. Model UltraSTEM 100). STEM imaging was performed at 60 kV to reduce the electron knock-on damage. The convergence angle is set to be ~30 mrad. All STEM images were acquired from ~86–200 mrad range. In operando XRD was measured using SmartLab$^®$ X-ray diffraction system (Rigaku Co.) with a battery cell attachment. XRD patterns were scanned by D/teX Ultra 250 detector from $2\theta = 3°$ to 40° with step size of 0.04° and scanning speed of 1° or 2° per minute under Bragg-Brentano focusing. Cu $K_α$ radiation ($\lambda = 1.5405$ Å) was used and the voltage and current was 40 kV and 44 mA, respectively. Ring-shaped Mg metal anode was placed on the Be window to avoid electrochemical dissolution of Be metal at >0.53 V vs Mg/Mg$^{2+}$. HE-XRD was carried out at beamline 11-ID-C of the Advanced Photon Source (APS) at Argonne National Laboratory (ANL). The wavelength of the X-ray was 0.10798 Å; such high-energy X-ray has a large penetration depth that allowed for the detection of structural changes of the bulk material. The TiS$_2$ powder was collected, put into kapton tube, and sealed by epoxy glue inside the glove box. A Perkin Elmer large area X-ray detector was used to collect the 2-dimensional diffraction patterns in transmission mode. The measured 2-dimensional diffraction patterns were calibrated using a standard CeO$_2$ sample and converted to 1-dimensional intensity vs the diffraction angle ($2\theta$) patterns using Fit2D software[75]. The diffraction peaks were assigned to lattice planes ($hkl$) by simulating the patterns of TiS$_2$ with the corresponding interlayer distance using PowderCell 2.4 software[76]. Mg $K$-edge NEXAFS experiment was performed on beamline 6.3.1.2 (ISAAC) at the Advanced Light Source, Lawrence Berkeley National Laboratory with both total electron yield (TEY) and total fluorescence yield (TFY) detection modes simultaneously. XAS spectra were energy calibrated by measuring MgO before and after the measurements. The NEXAFS measurements at S $K$-edge were performed at the Advanced Photon Source (APS) on the bending-magnet beamline 9-BM-B with electron energy of 7 GeV and average current of 100 mA. The radiation was monochromatized by a Si(111) double-crystal monochromator. Harmonic rejection was accomplished with Harmonic rejection mirror. All spectra of samples were collected in fluorescence mode by Vortex detector. For energy calibration, the peak position of sodium thiosulfate was adjusted to 2469.2 eV by Gaussian fitting. Data reduction and analysis were processed by Athena software. $^1$H-NMR samples were prepared by suspending thoroughly washed and dried samples in DMSO-$d_6$ by ultrasonication for 0.5 h and then heated at 70 °C for 1 h in a tight-sealed vial. $^1$H-NMR (400 MHz, DMSO-$d_6$, $\delta$) of PY14$^+$: 0.918 ($t$, 3H), 1.298 (m, 2H), 1.660 (m, 2H), 2.060 (m, 4H), 2.955 (s, 3H); of THF: 1.747 (quint, 4H), 3.589 ($t$, 4H). The composition for stage 3 was determined by combining thermogravimetric analysis (TGA), $^1$H-NMR, and ICP results. The gross value of weight percentage of PY14 and THF was obtained by TGA, because both of the organic species evaporate or completely decompose to gaseous products by 500 °C in an inert atmosphere[77]. Then the molar ratio of PY14 and THF ($x/y$) was measured by $^1$H-NMR and the molar ratio of Mg to Ti ($z$) was measured by ICP-OES.

**Data availability**. The data that support the findings of this study are available from the corresponding author upon request.

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

## Acknowledgements

Y.Y. acknowledges the financial support from the Office of Naval Research Young Investigator Award (No. N00014-13-1-0543), the National Science Foundation (CMMI-1400261), the TcSUH core funding, and the University of Houston Start-up Fund. J.L. and S.T.P. acknowledge support from the US Department of Energy (No. DE-FG02-09ER46554). S.T.P. also acknowledges support from the Gas Subcommittee Research and Development under Abu Dhabi National Oil Company (ADNOC). X.Q. and H.W. acknowledge the start-up funds from Texas A&M University, and portions of this research were conducted with the advanced computing resources provided by Texas A&M High Performance Research Computing. W.Z. acknowledges support by the Department of Energy Office of Science, Basic Energy Sciences, Materials Science and Engineering Directorate. The STEM characterization was supported in part through a user project supported by ORNL's Center for Nanophase Materials Sciences (CNMS), which is sponsored by the Scientific User Facilities Division, Office of Basic Energy Sciences of US Department of Energy. The Advanced Light Source is supported by the Director, Office of Science, Office of Basic Energy Sciences, of the U.S. Department of Energy under Contract No. DE-AC02-05CH11231. Use of the Advanced Photon Source (9-BM and 11-ID) was supported by the U.S. DOE, Office of Basic Energy Sciences, under Contract No. DE-AC0206CH11357.

## Author contributions

H.D.Y. and Y.Y. conceived the concept. H.D.Y. and H.D. conducted electrochemical measurements. J. Lin, W.Z. and S.T.P. conducted STEM and EELS experiments, and analysis. H.W. and X.Q. conducted first-principles calculations. J.G., Y. Liu, L.M., T.W., and J. Lu conducted the Mg and S K-edge measurements. Y. Li conducted XPS and ICP analysis. Q. R., Y. Li, and H.D. conducted EDS analysis. Y.J., H.D.Y. and Y. Liang conducted NMR analysis. H.D.Y. and Q.A. conducted in operando XRD measurement. Y.Y. supervised the research. H.D.Y., Y. Liang, and Y.Y. co-wrote the paper. All authors discussed the results and commented on the manuscript.

## Additional information

**Competing interests:** The authors declare no competing financial interests.

