## [Peer Review File · Nature Communications]

Reviewers' comments:

Reviewer #1 (Remarks to the Author):

The manuscript reports and investigates the intercalation and diffusion of $MgCl^+$ ion in a titanium disulfide cathode. The intercalation of rather a large ion was possible by the interlayer expansion of the cathode through cointercalation of a pyrrolidinium ion which comes from the ionic liquid added to the electrolyte solution. The authors claim a 239 mAh/g reversible capacity. There are several important core issues with this work: First, the concept and report of a faster diffusion owing to the lower polarity of $MgCl^+$ vs. Mg^{2+} is not a surprise and is in fact expected from simple chemical intuition (indeed its no surprise that the computations the authors report support this). Second, the authors "ignore" the presence of the large and heavy pyrrolidinium ions in the cathode structure when they report the 239 mAh/g which is incorrect and the capacity should be much lower due to these heavy ions lodging in the structure (also the presence of some solvent coordinated to $MgCl$ is ignored). The authors don't also report the volumetric capacity of this cathode and the effect of large volume expansion on lowering it. Last, the use of chlorides is not desired in Mg battery as they are corrosive. For the aforementioned reasons I don't recommend the publication of this article in Nature comm.

Reviewer #2 (Remarks to the Author):

This is an original and novel piece of work, in which a new idea of Mg ions intercalation cathode was described and demonstrated.

The idea of expanding TiS_2 , thus enabling intercalation of relatively stable $MgCl^+$ ions without the need of decomposing the monovalent ion. The performance presented and the analytical work described in the paper is fine. I did not find scientific/technical mistakes.

Before proceeding further, I would like the authors to answer the following questions, based on which it will be possible to evaluate the work.

1. To what extent, the expanded TiS_2 is stable in solutions? Do the organic cations remain there with no change?
 2. Is it possible to operate cells with the practical ratio between the electrodes surface and the solution volume?
 3. Can the authors comment about the de-solvation mechanism that strips the $MgCl^+$ ions from the rest of the solvation shell (complicated structures!) upon intercalation/de-intercalation processes?
 4. What is the effect of temperature? What is the appropriate temperature range in which these electrodes demonstrate a fully reversible behavior?.
 5. The paper shows one hundred cycles. This is too little for a rechargeable battery system!. Do the authors have data on more prolonged cycling?
 6. Is there any effect of the presence of IL moieties in these solutions on the reversible behavior of Mg electrodes in these solutions?
- End.

Reviewer #3 (Remarks to the Author):

This paper describes new approach for using Mg as transporting ions in rechargeable batteries that addresses major factors that limit their current application in Mg conventional battery cells. The authors unravel a new effective way of expanding interlayer distance in layered TiS_2 electrodes that enables unhindered diffusion of electroactive Mg complex species and diminish interaction of free Mg ions with exposed cathode atoms. The authors show in depth analysis of the mechanism of expanding of interlayer spacing during intercalation of pyrrolidinium based cation intercalation as

well as the change of the chemical nature of electrode states in various stages of intercalation/deintercalation. This detailed understanding of the elemental processes involved in MgCl^+ cycling in pillared TiS_2 structure enabled them to design an efficient Mg battery cell that shows remarkable reversibility, rate of cycling and Columbic efficiency for a Mg rechargeable battery. The use of XANES for understanding fine tuning of the coordination state of different Mg complexes is particularly useful for understanding the underlying mechanisms of intercalation and importance of using tetracoordinated Mg for building an efficient Mg cell.

However, the reversible Mg cell that the authors build appears to work only with 1 electron per Mg due to the charge of cycling with MgCl^+ complex. This should be addressed in the introduction. Also, although theoretical predictions suggest no difference between 10.9 and 18.6 Å, the authors find ~25% decrease in the capacity between the two electrode configurations. Is there steric hindrance of the intercalated pyrroliidinium cation? It was not clear also if pyrroliidinium cation contributes to the capacity. While on Page 6 the authors state that there is no intercalation of MgCl^+ at stage 1 (1 V), in Figure 5 at 1 V the cell shows almost half of the total capacity of Mg. It would be nice if the authors could obtain the shift in the Ti edge at different stages of cycling. Also I am little confused that the cell shows such a remarkable rate capability but stage 2 and stage 3 differ only in time while the potential stays the same when the capacity changes from 200 to 500 mAh g⁻¹ (Figure 3a). Also, in the figure 3b spacing in stage 2 appears to be larger than in stage 3 or 4.

In summary I highly recommend this paper to be published in Nature Communication after addressing of the comments raised in this review. The paper represents novel and distinguished contribution in the important field of developing new alternative rechargeable batteries

Reviewer #4 (Remarks to the Author):

This paper reports on the discovery of a novel mechanism of Mg intercalation involving MgCl^- dimolecules in layered intercalation compounds in which the layers are separated by the insertion of organic pillars (Py_{14}^+). This is a very important finding in Mg-battery research. The paper is very nicely written and combines a wide array of advanced experimental and computational approaches to paint a detailed picture of the relevant mechanisms that operate during the MgCl^- insertion and removal from expanded TiS_2 . The work is an important contribution to efforts devoted to discovering new high-density Mg cathode materials. It is a transformative contribution to this rapidly evolving field.

Before it can be published, however, the authors must address the following:

1. Diffusion of Mg and its sensitivity to the c-lattice parameter of layered TiS_2 was extensively studied in reference 51 – however the authors do not acknowledge this in the results section describing the theoretical modeling efforts. Ref 51 must be cited in that section.
2. The paper reports impressive capacities. These are calculated using the weight of TiS_2 and MgCl^- . However, the active electrode materials are not simply TiS_2 , but TiS_2 containing a high concentration of organic pillars. While it is fine to also mention capacities assuming only the weight of TiS_2 and MgCl^- for comparative purposes with past literature, the paper must also report capacities accounting for the actual weight of the active electrode material (i.e. TiS_2 with Py_{14}^+ and MgCl^-). Ultimately it is this weight that will be relevant if this chemistry and approach is to find itself in commercial batteries.
3. Supplementary figure 8 should be moved to the main text (e.g. as part of Figure 5 of the main text). The GITT measured voltage curves accurately show the true thermodynamic voltage profile for this electrode chemistry and for the reaction mechanism involving MgCl^- cations. Supplementary Figure 8 also nicely shows remnant polarization, which could be due to a hysteretic effect that may arise when a second mobile, yet more sluggish species is present (e.g. Py_{14}^+). Can the authors

comment on what seems to be 'thermodynamic' polarization in the GITT voltage profiles. A relevant reference in this context is: Yu, H.-C. et al., *Energy Environ. Sci.*, 7, 1760–1768, 2014.

4. Was the diffusion coefficient of Figure 5(f) measured during discharge or charge? This should be clarified in the text. The reduction in the diffusion coefficient with increasing concentration could be due to a divacancy diffusion mechanism as is well known to occur in Li-intercalation compounds. A relevant reference in this context is: Van der Ven, A.; Bhattacharya, J.; Belak, A. A. *Acc. Chem. Res.*, 46, 1216–1225, 2013

Response to Reviewers' Comments

We thank all of the four Reviewers for finding this work original and novel in Mg-battery research. We have significantly revised the manuscript with the guidance and inspiration of the constructive comments from these Reviewers. We apologize for the delay in responding, however, the additional measurements took a lot of time. We feel we have made progress in addressing the major points and hope this revised version of our manuscript could be suited for publication.

Reviewer #1

1. *The manuscript reports and investigates the intercalation and diffusion of MgCl⁺ ion in a titanium disulfide cathode. The intercalation of rather a large ion was possible by the interlayer expansion of the cathode through cointercalation of a pyrrolidinium ion which comes from the ionic liquid added to the electrolyte solution. The authors claim a 239 mAh/g reversible capacity. There are several important core issues with this work:*

First, the concept and report of a faster diffusion owing to the lower polarity of MgCl⁺ vs. Mg²⁺ is not a surprise and is in fact expected from simple chemical intuition (indeed its no surprise that the computations the authors report support this).

Our response: We would like to clarify that this work is intended not as a surprise finding that is contradictory to chemical intuition, but as a discovery of an unprecedented battery chemistry started from an educated hypothesis and realized by rational design. The faster diffusion owing to the lower polarity of MgCl⁺ vs Mg²⁺ is predictable based on how polarization strength is defined. However, there is neither prior attempt nor theoretical prediction to use MgCl⁺ as a charge carrier for Mg batteries. The computation provides us with the necessary quantitative information on the diffusivity as a function of interlayer distance and the chemical structure of the ions, as well as the level of TiS₂ expansion required for Mg²⁺ and MgCl⁺ to get maximum diffusivities. While chemical intuition leads us to anticipate fast diffusion of MgCl⁺ than Mg²⁺, it was through the systematic computational studies that we acquired quantitative information to make this battery chemistry a reality.

In revision, we have included the rationalization behind choosing MgCl⁺ as the charge carrier at page 3 (also see our response to Reviewer #3's comment 2). We further discussed the necessity of the computational studies at page 4: "Although the faster diffusion owing to the lower polarity of MgCl⁺ vs Mg²⁺ is expectable based on the predictably decreased polarization strength, computational studies provide quantitative information on the diffusivity as a function of interlayer distance and the chemical structure of the ions, as well as the extent of TiS₂ expansion required for Mg²⁺ and MgCl⁺ to achieve maximum diffusivities."

2. *Second, the authors "ignore" the presence of the large and heavy pyrrolidinium ions in the cathode structure when they report the 239 mAh/g which is incorrect and the capacity should be much lower due to these heavy ions lodging in the structure (also the presence of some solvent coordinated to MgCl is ignored).*

Our response: We agree with the Reviewer that the specific capacity values based on the mass of TiS_2 and that of the whole composite are both important for evaluation of the materials/chemistry, and we should include both values in the manuscript. We have determined the composition of $(\text{MgCl})_z\text{TiS}_2[(\text{PY14})_x(\text{THF})_y]$ at stage **3** by combining TGA, NMR, and ICP results (detailed in the revised manuscript). Based on the composition of $(\text{MgCl})_{1.0}\text{TiS}_2[(\text{PY14})_{0.20}(\text{THF})_{0.16}]$ at stage **3** (*i.e.* with the mass of PY14^+ and THF taken into account), the specific capacity of the cathode is 173 mAh/g. When we tested the cell at elevated temperatures in response to Reviewer #2, we found that up to 1.7MgCl^+ can be reversibly intercalated per unit TiS_2 , giving rise to a specific capacity of 400 mAh/g based on the mass of TiS_2 (Figure 6c), or 269 mAh/g based on the total mass of TiS_2 , PY14, and THF. These values are now added to the revised manuscript. Note that our originally reported specific capacity of 239 mAh/g is in line with these additional values. The specific capacity can be further enhanced by engineering more light-weight pillaring molecules.

In revision, we have addressed the composition of exTiS_2 at page 8: “Combining these results and thermogravimetric analysis (Supplementary Fig. 7 and Supplementary Table 4), we obtain the composition of the discharged compound at stage 3 as $(\text{MgCl})_{1.0}\text{TiS}_2[(\text{PY14})_{0.20}(\text{THF})_{0.16}]$.” The experimental detail is included in the Methods section: “The composition for stage **3** was determined by combining thermogravimetric analysis (TGA), $^1\text{H-NMR}$, and ICP results. The gross value of weight percentage of PY14 and THF was obtained by TGA, because both of the organic species evaporate or completely decompose to gaseous products by $500\text{ }^\circ\text{C}$ in an inert atmosphere⁷⁷. Then the molar ratio of PY14 and THF (x/y) was measured by $^1\text{H-NMR}$ and the molar ratio of Mg to Ti (z) was measured by ICP-OES.”

The capacities based on the mass of TiS_2 and the composite exTiS_2 , which is $\text{TiS}_2[(\text{PY14})_{0.20}(\text{THF})_{0.16}]$, are both reported: “ exTiS_2 shows a high reversible capacity of 239 mAh g^{-1} based on the mass of TiS_2 at the current density of $24 \text{mA g}_{\text{TiS}_2}^{-1}$ (0.1C-rate), or 173 mAh g^{-1} based on the composite mass of $\text{TiS}_2[(\text{PY14})_{0.20}(\text{THF})_{0.16}]$ at room temperature (Fig. 5a)” (page 9) and “At $60\text{ }^\circ\text{C}$, the cell reaches a capacity of 400 mAh $\text{g}_{\text{TiS}_2}^{-1}$ (394Ah L^{-1}), or 269 mAh g^{-1} based on the mass of the composite, corresponding to the intercalation of 1.7MgCl^+ per unit TiS_2 ” (page 11). The potential to increase capacity by engineering the pillars is discussed (page 12): “Further engineering of less bulky and more light-weight pillar species may require smaller structural adjustments and also lead to higher specific capacity (which is determined by the total mass of the host and the pillar).” In the abstract, it is made clear that the reported capacity is based on TiS_2 : “corresponding to up to 400 mAh g^{-1} capacity based on the weight of TiS_2 ”. The figure (Supplementary Figure 7) and the table (Supplementary Table 4) related to TGA measurements are included in the revised Supplementary Information. At the same time, Figures 5 now explicitly denotes that the capacity is calculated based on the mass of TiS_2 ($\text{g}_{\text{TiS}_2}^{-1}$) to avoid confusion.

Supplementary Figure 7. TGA for the samples of stages **0** and **3** in nitrogen flow at the heating rate of $5\text{ }^{\circ}\text{C min}^{-1}$.

Supplementary Table 4. TGA measurements and the compositions of stages **0** and **3** that were derived by combining TGA, ICP-OES, EDS, and NMR results

Stage	Weight (%) at 500 °C	Derived composition
0	82.9	TiS_2
3	55.1	$(\text{MgCl})_{1.0}\text{TiS}_2[(\text{PY14})_{0.20}(\text{THF})_{0.16}]$

3. *The authors don't also report the volumetric capacity of this cathode and the effect of large volume expansion on lowering it.*

Our response: We thank the reviewer for pointing out that the interlayer expansion may lower the volumetric capacity. The volumetric capacity of exTiS_2 is 235 Ah/L, which is 3.6 times as high as that of pristine TiS_2 (66 Ah/L). In the revision, we found that the volumetric capacity of exTiS_2 reaches 394 Ah/L at 60 °C based on a 400 mAh/g capacity. The volumetric capacity can be further enhanced by optimizing the level of expansion using pillars with different structures. For comparison, the volumetric capacity for Chevrel-phase Mo_6S_8 is 519 Ah/L based on a 100 mAh/g specific capacity.

The enhanced volumetric capacity compared with the pristine material justifies the interlayer expansion. It is noteworthy that other strategies such as synthesizing mesoporous or composite materials also lower the volumetric capacity, but they have been utilized because of the prominent benefits from the enhanced kinetics despite the lowered volumetric capacity.

In revision, we have included the volumetric capacity values as Supplementary Table 5 and the corresponding discussion on page 9: “The volumetric capacity of exTiS_2 is 235 Ah L^{-1} , which is 3.6 times as high as that of pristine TiS_2 (66 Ah L^{-1}) but 55% lower than that of Mo_6S_8 (519 Ah L^{-1}) due to the decreased density caused by the volume expansion (Supplementary Table 5)” and “At 60 °C, the cell reaches a capacity of 400 mAh $\text{g}_{\text{TiS}_2}^{-1}$ (394 Ah L^{-1}),”

Supplementary Table 5. Gravimetric and volumetric capacity of pristine TiS₂, exTiS₂, and Chevrel phase Mo₆S₈ at 25 °C

	Pristine TiS ₂	exTiS ₂	Chevrel Mo ₆ S ₈
Specific capacity (mAh g ⁻¹)	20.5	173	100
Volumetric capacity (Ah L ⁻¹)	66	235	519

4. *Last, the use of chlorides is not desired in Mg battery as they are corrosive.*

Our response: We agree with the reviewer that the corrosion induced by chlorides can be a potential problem for Mg batteries. However, we note that such corrosion does not happen until the voltage reaches a high value. For example, stainless steel is stable towards chlorides up to 2.2 V vs Mg/Mg²⁺ in APC electrolyte. The anodic stability window can be further raised up to 3 V vs Mg/Mg²⁺ by using alternative current collectors such as graphite and molybdenum^{1,2}. For this reason, corrosion is not an issue in the battery system reported herein where we only need to charge to 2 V vs Mg/Mg²⁺. On the anode side, chlorides provide effective protection on Mg anodes for the reversible Mg deposition and stripping³⁻⁵. Based on such electrochemical properties of chloride-based electrolytes, more than 2,000 and 7,000 cycles have been demonstrated for Mg and Al rechargeable batteries, respectively. Therefore, chloride-containing electrolytes continue to receive wide interests for Mg batteries.

Recently, Oh *et al.* have reported that PY14Cl, the same ionic salt that we used in our work, is an effective inhibitor against the corrosion by Cl⁻ at high electrode potentials when it was added in a Mg ion electrolyte⁶. Therefore, PY14⁺ is not only a pillar for expanding TiS₂ but also an effective inhibitor for possible corrosion by Cl⁻ in the electrolyte.

For even higher voltage windows, halogen-free electrolytes are under development⁷. In those cases, the anion (A⁻) can associate with Mg²⁺ to give MgA⁺ species, which can be used as the active diffusive species in expanded materials. In this sense, the present study provides a general guideline and design principles for the intercalation of such generalized Mg complex ions into expanded interlayers.

In revision, we have included the above discussion at page 14: “Recent years have seen increasing concerns about potential corrosion problems related to chloride-containing electrolytes. Oh *et al.* have reported that PY14Cl is an effective inhibitor against the corrosion by Cl⁻ at high potentials⁶. Therefore, PY14⁺ is not only a pillar for expanding TiS₂ but also a corrosion inhibitive additive. Meanwhile, halogen-free electrolytes are under development for even wider voltage windows²⁴. In those cases, the anion (A⁻) can associate with Mg²⁺ to form MgA⁺ ions which can be used as the charge carrier in expanded materials. In this sense, the present study provides general guidelines and design principles for the intercalation of such generalized Mg complex ions into expanded interlayers.”

5. *For the aforementioned reasons I don't recommend the publication of this article in Nature comm.*

Our response: We hope our additional data analysis and clarification convinces the reviewer

to make a more positive decision on our work.

Reviewer #2

1. *This is an original and novel piece of work, in which a new idea of Mg ions intercalation cathode was described and demonstrated.*

The idea of expanding TiS₂, thus enabling intercalation of relatively stable MgCl⁺ ions without the need of decomposing the monovalent ion. The performance presented and the analytical work described in the paper is fine. I did not find scientific/technical mistakes. Before proceeding further, I would like the authors to answer the following questions, based on which it will be possible to evaluate the work.

Our response: We thank the Reviewer for judging our work “original and novel”.

2. *To what extent, the expanded TiS₂ is stable in solutions? Do the organic cations remain there with no change?*

Our response: Our results indicate sufficient stability of the organic cation-pillared, expanded TiS₂ composite. The organic cation PY14⁺ itself is known to be electrochemically stable at potentials as low as -0.4 V vs Mg/Mg²⁺⁸. The FE-SEM images of the expanded TiS₂ electrodes at different stages (0-5) evidence the structural integrity of the expanded layers during the cycling (Supplementary Fig. 5). And the *in-situ* XRD results in Figure 3a and Supplementary Fig. 4 show that the expanded structure does not change upon charging or repeated cycling. More specifically, the elemental mapping of the fully charged, deintercalated TiS₂ at stage 4 (in Figure 3d) confirms that the organic cations, which are represented as carbon (C), remain in the interlayer of TiS₂. The latter observation has been mentioned in the manuscript on page 7 as “Cross-sectional elemental mapping at stage 4 shows alternating layers of Ti and C, which is a clear evidence that organic PY14⁺ “pillars” stay in the van der Waals gap of TiS₂ after the expansion (Fig. 3d). The *ex*TiS₂ electrodes remain compact without exfoliation during the cycling (Supplementary Fig. 5).”

The long-term integrity of the expanded composite is reflected by the cycling stability of the PY14⁺-pillared TiS₂ in pure APC electrolyte (*i.e.* without PY14⁺); 80% capacity retention after 350 cycles, similar to the cyclability observed in the PY14⁺-containing electrolyte (Supplementary Figure 11). This result reaffirms that the organic cations are chemically stable and sufficiently immobile and stay in the structure with no change.

Supplementary Figure 11. Cycling stability of PY14⁺-pillared, expanded TiS₂ in APC electrolyte without PY14⁺ ions at 1C-rate (2.0–0.1 V). The cycling retention was 80% of the initial capacity after 350 cycles.

In revision, the Results section on page 10 reads “Supplementary Fig. 11 shows stable cycling of 80% capacity retention after 350 cycles, similar to the cyclability observed in the PY14⁺-containing electrolyte. This result reaffirms that the organic cations are chemically stable and sufficiently immobile and stay in the structure with no change during the cycling.”

3. *Is it possible to operate cells with the practical ratio between the electrodes surface and the solution volume?*

Our response: We thank the reviewer for the practically important comment. As the MgCl⁺ battery chemistry utilizes chlorine moieties from the electrolyte in addition to Mg moieties from the Mg metal anode, the volume of electrolyte is an important factor for the practical concern. Herein we calculated the necessary volume of electrolyte per unit surface area (1 cm²) of electrode with areal capacity of 1 mAh cm⁻², which approximates a practical scenario. The volume of electrolyte is determined by the amount of MgCl₂ needed to sustain the reaction (equation (3) from the original submission):

Using a concentration of 1.0 M for MgCl₂⁹, the necessary volume of electrolyte is (1 mAh * 3.6 C/mAh) / [2 * 96485 C/mol * 1.0 mol/L] = 18.7 μL per unit area of 1 cm². This amount of electrolyte can be accommodated by the separator alone: a commercial separator with 300 μm thickness and 80% porosity can uptake 24 μL of electrolyte per 1 cm². Also note that the cation concentration of the electrolyte (hence the conductivity) stays unchanged throughout the cell operation. Therefore, it is possible to operate cells with the practical ratio between the electrode surface and the solution volume.

In revision, we have included the above analysis at page 13: “According to Equation (3), two moles of (MgCl)TiS₂ are formed with the consumption of one mole of MgCl₂.”

Considering a 1.0 M concentration for MgCl_2 ⁶², the necessary volume of electrolyte to match an areal capacity of 1 mAh cm^{-2} (which approximates the areal capacity of a practical cell) is $18.7 \mu\text{L cm}^{-2}$. This amount of electrolyte is sufficiently small to be accommodated in practical batteries: commercial separators with $300 \mu\text{m}$ thickness and 80% porosity may uptake $24 \mu\text{L cm}^{-2}$ of electrolyte⁶³.”

4. *Can the authors comment about the de-solvation mechanism that strips the MgCl^+ ions from the rest of the solvation shell (complicated structures!) upon intercalation/de-intercalation processes?*

Our response: We thank the reviewer for the suggestion to elaborate on the de-solvation mechanism of MgCl^+ from the rest of the complicated solvation shell. The MgCl^+ ion is known to prefer tetra-coordination, so it is solvated by three THF ligands to form $\text{MgCl}^+\cdot 3\text{THF}$. The THF ligands coordinate to Mg via cation-dipole interactions, which are generally weaker than the cation-anion interactions between Mg^{2+} and Cl^- . From calculations, the dissociation energy for Cl and THF from Mg is 3.0 and 0.8 eV, respectively¹⁰, which makes the dissociation of THF $\sim 10^{37}$ times more likely than that of Cl. And the THF ligands will be (comparatively) easily stripped. On the electrode-electrolyte interface, the partially stripped $\text{MgCl}^+\cdot(3-n)\text{THF}$ will be coordinated by S^{2-} anions of TiS_2 and form tetra-coordinated $\text{MgCl}^+\cdot(3-n)\text{THF}\cdot n\text{S}^{2-}$. As this process repeats, MgCl^+ will be finally coordinated by three S as illustrated in Figure 1b.

In revision, we have discussed the relative ease of desolvation (page 3): “only low-energy desolvation ($E_a \sim 0.8 \text{ eV}$) but not high-energy Mg–Cl scission ($E_a > 3 \text{ eV}$) is necessary before intercalation”.

5. *What is the effect of temperature? What is the appropriate temperature range in which these electrodes demonstrate a fully reversible behavior?*

Our response: We thank the reviewer for pointing out the important factor of temperature. In response to the reviewer’s question, we tested the $\text{exTiS}_2 \mid \text{Mg}$ cells at different temperatures. As the temperature increased from -45 to $60 \text{ }^\circ\text{C}$, we observed significant increase in MgCl^+ intercalation capacity (new Figure 5e). This improvement can be attributed to increased MgCl^+ diffusivity; considering the migration barrier of 0.18 eV for MgCl^+ in expanded TiS_2 , the diffusivity increases to 209% when the temperature increases from 25 to $60 \text{ }^\circ\text{C}$ according to the Arrhenius relation with temperature ($D \propto e^{-E_a/kT}$).

We are excited to report that at $60 \text{ }^\circ\text{C}$, the cell reaches a capacity of $400 \text{ mAh g}_{\text{TiS}_2}^{-1}$, or 1.67 MgCl^+ per unit TiS_2 . Having more than one electron reversibly stored per TiS_2 formula is to our knowledge unprecedented. Each 1T- TiS_2 unit possesses two distinguishable sites: the octahedral (Ti top) and tetrahedral (hollow top) sites. Without interlayer expansion, only one site is energetically viable for ion intercalation (octahedral site in this case). Intercalation of a second monovalent ion only happens at exceedingly low potential (-0.27 V vs Mg/Mg^{2+}) and the process is not reversible (see the new Supplementary Figure 12). When TiS_2 is expanded, intercalation at both sites exhibits similar energy levels per our simulation results (Figure 2b), thereby enabling more than one MgCl^+ to intercalate per unit TiS_2 .

We did not test the cells at temperatures higher than $60 \text{ }^\circ\text{C}$ because of the limitation of the THF solvent (boiling point: $66 \text{ }^\circ\text{C}$). However, further increase in the operational temperature

may be possible by engineering the electrolyte, such as substituting solvents having much higher boiling points such as glymes for THF.

In summary, the MgCl^+ intercalation into $ex\text{TiS}_2$ is fully reversible over a wide temperature range from -45 to 60 °C. Higher temperature leads to higher MgCl^+ diffusivity and nearly doubled reversible capacity.

Figure 5e. Voltage profiles of $ex\text{TiS}_2$ electrodes at temperatures varied from -45 to 60 °C.

Supplementary Figure 12. Voltage profiles for Li intercalation in pristine TiS_2 with the discharge cutoff set to 1.5 (red) and 0.3 (black) V vs Li/Li^+ . The lowered cutoff voltage leads to more than one electron transfer per unit TiS_2 but with poor reversibility.

In revision, we have included the above figures (Figure 5e and Supplementary Figure 12) with corresponding discussion right before the Discussion section (page 11).

6. *The paper shows one hundred cycles. This is too little for a rechargeable battery system!. Do the authors have data on more prolonged cycling?*

Our response: We thank the reviewer for the comment. More prolonged cycling up to 500 cycles are shown below in revised Figure 5b. The cycling retention was about 80% of the initial capacity after 400 cycles at 1C-rate.

Figure 5b. Cycling retention of $exTiS_2$ | Mg full cell up to 500 cycles at 1C-rate. The dip of capacity at around 150th cycle was due to the lowered room temperature caused by the temporary failure of air-conditioning system.

In revision, the above figure for prolonged cycling has been included as Fig. 5b. The corresponding discussion now reads “In terms of cycling stability in Fig. 5b, the $exTiS_2$ electrode exhibits 80% capacity retention after 400 cycles at 1C-rate with coulombic efficiency consistently higher than 99% (Fig. 5b).”

7. *Is there any effect of the presence of IL moieties in these solutions on the reversible behavior of Mg electrodes in these solutions?*

Our response: The presence of IL moieties slightly modifies the reversibility and kinetics of Mg electrodes, which information was included as Supplementary Table 1 and Supplementary Figure 2 in the original submission. We have now included detailed discussion on these effects in the revised main text in manuscript. In fact, we chose the pyrrolidinium-based ionic liquid (IL) because of its excellent cathodic stability compared to other ILs. The PY14⁺ ion has been reported to be stable down to -0.4 V vs Mg/Mg²⁺ (Ref. ⁸). Supplementary Figure 2 (see below) compares the electrolyte solutions with (solid) and without (dashed) PY14⁺. Both electrolyte solutions allow for reversible Mg deposition and dissolution. The overpotential for Mg deposition and dissolution very slightly increases (from 0.14 to 0.16 V) and noticeably decreases (from 0.86 to 0.67 V), respectively, with the addition of 0.2 M PY14Cl to APC. The coulombic efficiency slightly decreases from 99.9% to 95.2%, which can be improved by optimizing the concentration of PY14Cl based on a recent report by Oh *et al.*, where the Mg electrode achieves a 99% efficiency in the presence of PY14Cl⁶.

In revision, we have included the discussion at page 6 “We chose chemically stable 1-butyl-1-methylpyrrolidinium ion (PY14⁺) as an organic “pillar”^{48,49}, which expands TiS_2 layers *in situ*, i.e. by discharging a complete TiS_2 /Mg cell using the electrolyte containing PY14⁺ ions. The reversible Mg deposition and dissolution in the electrolyte solution is not hampered with the addition of the PY14⁺ ions (Supplementary Table 1 and Supplementary Fig. 2).”

Reviewer #3

1. *This paper describes new approach for using Mg as transporting ions in rechargeable batteries that addresses major factors that limit their current application in Mg conventional battery cells. The authors unravel a new effective way of expanding interlayer distance in layered TiS₂ electrodes that enables unhindered diffusion of electroactive Mg complex species and diminish interaction of free Mg ions with exposed cathode atoms. The authors show in depth analysis of the mechanism of expanding of interlayer spacing during intercalation of pyrroliidinium based cation intercalation as well as the change of the chemical nature of electrode states in various stages of intercalation/deintercalation. This detailed understanding of the elemental processes involved in MgCl⁺ cycling in pillared TiS₂ structure enabled them to design an efficient Mg battery cell that shows remarkable reversibility, rate of cycling and Columbic efficiency for a Mg rechargeable battery. The use of XANES for understanding fine tuning of the coordination state of different Mg complexes is particularly useful for understanding the underlying mechanisms of intercalation and importance of using tetracoordinated Mg for building an efficient Mg cell.*

Our response: We thank the Reviewer for the positive comments on the novelty and the key aspects of this work.

2. *However, the reversible Mg cell that the authors build appears to work only with 1 electron per Mg due to the charge of cycling with MgCl⁺ complex. This should be addressed in the introduction.*

Our response: We agree with the Reviewer that we need to better address the monovalent nature of the MgCl⁺ ion and its implication on the battery chemistry reported in this work. Changing the divalent Mg²⁺ to the monovalent MgCl⁺ as the charge carrier make Mg ions similar to alkaline metal ions where one electron is transferred per metal atom. The reduced charge number (from 2 to 1) and increased ion radius (from 0.72 to ~4.6 Å) leads to a ~80x decrease in polarization strength (which is proportional to the charge number and inversely proportional to the square of the ionic radius) of Mg ion, and hence much faster diffusion.

In revision, we included the rationalization on page 3: “Moving from the divalent Mg²⁺ to the monovalent MgCl⁺ as the charge carrier makes Mg ions similar to one-electron-transfer alkaline metal ions where (1) only low-energy desolvation ($E_a \sim 0.8$ eV) but not high-energy Mg–Cl scission ($E_a > 3$ eV) is necessary before intercalation and (2) the polarization strength of the ion, and hence the ion diffusion energy barrier, is low (Fig. 1b)”.

3. *Also, although theoretical predictions suggest no difference between 10.9 and 18.6 Å, the authors find ~25% decrease in the capacity between the two electrode configurations. Is there steric hindrance of the intercalated pyrroliidinium cation?*

Our response: We would like to make sure we understood the comment correctly. We think the Reviewer is referring to the comparison of *ex*TiS₂ before (stage 2) and after (stage 3) activation at low potential: the former shows a capacity of 60 mAh/g, or ~25% of that of the latter (239 mAh/g). The different performance between the two *ex*TiS₂ is likely due to the rupturing structure of the latter rather than spacing and steric hindrance of the PY14⁺ cation, because both samples have an interlayer spacing of ~18.6 Å and the same amount of

intercalated PY14⁺. Only the sample at stage 1 has an interlayer spacing of 10.9 Å, which has not been subjected to Mg cell tests. We agree with the Reviewer that the steric hindrance of the intercalated PY14⁺ will affect the electrochemical performance of the composites, which was not considered in our simulation. This may be the reason why fast diffusion of MgCl⁺ requires structural adjustments beyond interlayer spacing of 10.9 Å (e.g., interlayer expansion to 18.6 Å, intralayer ruptures as shown in Fig. 3b, etc.) A proper method to probe this is by altering the pillaring species.

In revision, we have discussed the possible steric effect of PY14⁺ at page 12: “It is noteworthy that the theoretical predictions suggest no difference in the diffusivity of MgCl⁺ as long as the interlayer distance is larger than 10.9 Å. However, the theoretical modeling does not account for the steric hindrance of the intercalated PY14⁺ cations. Therefore, in reality, fast diffusion of MgCl⁺ may require more substantial structural adjustments (e.g., interlayer expansion to 18.6 Å, intralayer ruptures as shown in Fig. 3b, etc.), which make the structure more accessible to MgCl⁺ ions. Further engineering of less bulky and more light-weight pillar species may require smaller structural adjustments and also lead to higher specific capacity (which is determined by the total mass of the host and the pillar).”

4. *It was not clear also if pyrroliidinium cation contributes to the capacity. While on Page 6 the authors state that there is no intercalation of MgCl⁺ at stage 1 (1 V), in Figure 5 at 1 V the cell shows almost half of the total capacity of Mg.*

Our response: We thank the Reviewer for pointing out the potentially confusing information in the manuscript. The pyrrolidinium ion (PY14⁺) does not contribute to the reversible capacity shown in Figure 5a, as we have addressed on page 8 in the original submission where *ex*TiS₂ was tested in an electrolyte without PY14⁺. It does induce an irreversible capacity (~50 mAh/g) that only occurs during the activation process, i.e. the first discharge shown in Figure 3a and discussed on page 6. We have clarified the difference between the first discharge and following cycles in the revised manuscript.

In revision, to avoid misunderstanding, we have changed the caption of Fig. 3 as “Structural characterizations of TiS₂ during the initial activation.” To differentiate the TiS₂ before and after interlayer expansion, we use the term “*ex*TiS₂” exclusively for the activated/expanded sample (defined at page 7). We have also included a clarification of the contribution of PY14⁺ to capacities (page 7) “The initial activation is complete at this stage with PY14⁺ contributing a one-time irreversible capacity of ~50 mAh g⁻¹. P14⁺ do not contribute to the reversible capacity in the following cycles.”

5. *It would be nice if the authors could obtain the shift in the Ti edge at different stages of cycling.*

Our response: We thank the reviewer for the valuable suggestion. We prepared three activated samples at different stages of cycling, i.e., pristine, (MgCl)_{0.5}-TiS₂, (MgCl)_{1.0}-TiS₂. Unfortunately, we could not obtain the beamtime access for the Ti *K*-edge near-edge X-ray absorption fine structure (NEXAFS) measurement. However, we did manage to obtain the S *K*-edge NEXAFS measurement through collaboration with Dr. Jun Lu at Argonne National Lab using the beamline 9-BM-B at the Advanced Photon Source. S *K*-edge measurement could be more important to reflect the changes in the sulfur coordination environment as well

as the electronic states of Ti 3d orbitals since MgCl^+ directly coordinates to sulfur. The spectra were calibrated by sodium thiosulfate with the peak position at 2469.2 eV.

The new Figure 4e shows the S K-edge NEXAFS spectra of $(\text{MgCl})_x\text{TiS}_2$ for $x = 0.0, 0.5,$ and 1.0 . The NEXAFS spectra are displayed in Fig. 4e after background subtraction and normalization. Three S K-edge peaks C, C', and D represent the transitions from S 1s to S 3p orbitals. The C and C' peaks can be assigned to t_{2g} and e_g states from the hybridization of S 3p and Ti 3d orbitals via π^* and σ^* antibonding, respectively. The intensity and width of C and C' peaks decrease upon MgCl^+ intercalation, while peak C exhibits more pronounced decrease in the intensity than C'; but no noticeable energy shift is observed for the two peaks. Peak D, which can be assigned to hybridized S 3p and Ti 4s and 4p orbitals, shows a progressive shift toward lower energy and an increase in the intensity upon MgCl^+ intercalation. The observed spectral changes are similar to the experimental and theoretical study for Li^+ intercalation into TiS_2 ¹¹, in which the reduced intensity of C and C' peaks was originated from the structural distortion, with more pronounced influence on peak C due to the partial filling of the t_{2g} states by the charge transfer upon intercalation of ions. In our case, the structural distortion and charge transfer would be results from the MgCl^+ intercalation. The change of peak D may reflect the bonding between Mg and S atoms and the coordination number of S changes from three in TiS_2 to six in $(\text{MgCl})_x\text{exTiS}_2$, whereby the hybridization of S increases.

Figure 4e. Experimental XANES spectra at S K-edge for TiS_2 (black), $\text{MgCl}_{0.5}\text{-TiS}_2$ (blue, dashed), $\text{MgCl}_{1.0}\text{-TiS}_2$ (red).

In revision, we have added the above discussion in the main text (page 8) and updated the experimental information in the Supplementary Information. “The NEXAFS measurements at S K-edge were performed at the Advanced Photon Source (APS) on the bending-magnet beamline 9-BM-B with electron energy of 7 GeV and average current of 100 mA. The radiation was monochromatized by a Si (111) double-crystal monochromator. Harmonic rejection was accomplished with Harmonic rejection mirror. All spectra of samples were collected in fluorescence mode by Vortex detector. For energy calibration, the peak position of sodium thiosulfate was adjusted to 2469.2 eV by Gaussian fitting. Data reduction and analysis were processed by Athena software.”

6. Also I am little confused that the cell shows such a remarkable rate capability but stage 2 and stage 3 differ only in time while the potential stays the same when the capacity changes from 200 to 500 mAh g⁻¹ (Figure 3a).

Our response: This is another potentially confusing point that we certainly need to better explain. The transition from stage 2 to 3 shown in Figure 3a corresponds to the activation process that occurs during the first discharge only. The process is slow and shows a potential profile that is different from the following cycles which show remarkable rate capability. We have better explained the unique behavior of the activation process in the revised manuscript.

In revision, we have clarified in the main text that the *in operando* experiment was carried out during the initial activation process (page 6): “*In operando* X-ray diffraction (XRD) shed light on the structural evolution of TiS₂ during the initial activation of the cell”. We have changed the caption of Fig. 3 as “Structural characterizations of TiS₂ during the initial activation.”

7. Also, in the figure 3b spacing in stage 2 appears to be larger than in stage 3 or 4.

Our response: Our apology for the confusion. In the original submission (Fig. 2b), the scale bar for stage 2 in the TEM image is slightly longer than those for stage 3 and 4, hence leading to the false impression that the spacing at stage 2 appears to be larger.

In revision, to avoid misimpression, we have unified the scale bars for stages 2-4 to the same length. Now it is clear that stages 2-4 show similar interlayer distance on average but broader distribution in stage 3-4 due to the structural disorder.

Figure 3b. STEM images of stages 2 to 4 with unified scale.

8. In summary I highly recommend this paper to be published in Nature Communication after addressing of the comments raised in this review. The paper represents novel and distinguished contribution in the important field of developing new alternative rechargeable batteries

Our response: We are grateful for the reviewer’s recommendation of this work as novel, distinguished, and important contribution in the field.

Reviewer #4

1. *This paper reports on the discovery of a novel mechanism of Mg intercalation involving MgCl- dimolecules in layered intercalation compounds in which the layers are separated by the insertion of organic pillars (Py14+). This is a very important finding in Mg-battery research. The paper is very nicely written and combines a wide array of advanced experimental and computational approaches to paint a detailed picture of the relevant mechanisms that operate during the MgCl insertion and removal from expanded TiS₂. The work is an important contribution to efforts devoted to discovering new high-density Mg cathode materials. It is a transformative contribution to this rapidly evolving field. Before it can be published, however, the authors must address the following:*

Our response: We thank the Reviewer for the appreciation and judging, as a major expert in the field, our work as “a transformative contribution.”

2. *Diffusion of Mg and its sensitivity to the c-lattice parameter of layered TiS₂ was extensively studied in reference 51 – however the authors do not acknowledge this in the results section describing the theoretical modeling efforts. Ref 51 must be cited in that section.*

Our response: We agree with the Reviewer that Ref. 51 should be moved to the theoretical modeling section. While we placed ref. 51 in the modeling section in our earlier version of this manuscript, it was somehow moved to the current place without us noticing. Our apology for the negligence.

In revision, we have cited Ref. 51 in the first paragraph of Results (page 4) as the new Ref. 41: “The diffusion of Mg²⁺ in layered TiS₂ and its sensitivity to the interlayer spacing (c) has been extensively studied by previous theoretical modeling effort⁴¹, which evidenced significant decrease of the migration barrier by up to 10% expansion from the pristine state (i.e., from 5.7 to 6.3 Å). Herein we study the effect of further expansion of TiS₂ on the mobility of Mg²⁺ and MgCl⁺. As c increases from 5.7 to 10.9 Å, the Mg²⁺ migration barrier reduces from 1.06 to 0.51 eV (Fig. 2a,b) as a result of smaller total binding energy between Mg and S in TiS₂, in excellent agreement with the previous work⁴¹.”

3. *The paper reports impressive capacities. These are calculated using the weight of TiS₂ and MgCl. However, the active electrode materials are not simply TiS₂, but TiS₂ containing a high concentration of organic pillars. While it is fine to also mention capacities assuming only the weight of TiS₂ and MgCl for comparative purposes with past literature, the paper must also report capacities accounting for the actual weight of the active electrode material (i.e. TiS₂ with Py14+ and MgCl). Ultimately it is this weight that will be relevant if this chemistry and approach is to find itself in commercial batteries.*

Our response: We appreciate the Reviewer’s suggestion to also report specific capacity based on the mass of the entire composite, which coincides with Reviewer #1’s comment 2. Please see our response and revision to Reviewer #1’s comment 2. We have included in the revised manuscript the exact chemical composition of the expanded composite and the specific capacity based on the total mass. We have also clarified the figures when the reported values are based on the mass of TiS₂ only.

4. *Supplementary figure 8 should be moved to the main text (e.g. as part of Figure 5 of the main text). The GITT measured voltage curves accurately show the true thermodynamic voltage profile for this electrode chemistry and for the reaction mechanism involving MgCl cations. Supplementary Figure 8 also nicely shows remnant polarization, which could be due to a hysteretic effect that may arise when a second mobile, yet more sluggish species is present (e.g. Py14+). Can the authors comment on what seems to be ‘thermodynamic’ polarization in the GITT voltage profiles. A relevant reference in this context is: Yu, H.-C. et al., Energy Environ. Sci., 7, 1760–1768, 2014.*

Our response: Another excellent suggestion. We agree with the Reviewer that GITT curves contain thermodynamic information that reveals non-dissipative hysteresis. In the suggested reference, Van der Ven’s group examined the intrinsic thermodynamic and kinetic properties that are responsible for the large hysteresis in high capacity electrodes. In our GITT voltage curve (Figure 5d), the open circuit potential (denoted as red) during the cycling corresponds to the true thermodynamic voltage profile. There is a voltage gap between charge and discharge. According to Yu, H.-C. *et al.*¹², this can happen when the reaction mechanisms involve the redistribution of a second mobile yet much more sluggish species (PY14⁺ in this case). We have rearranged the figures and included the above discussion and relevant references in the revised manuscript.

In revision, we have moved the GITT curve to the main text as Figure 5d and included the discussion “The open circuit potential (denoted as red) during the cycling corresponds to the true thermodynamic voltage profile. There is a voltage gap between charge and discharge, reflecting a MgCl⁺ (de)intercalation mechanism that involves the redistribution of a second mobile yet much more sluggish species, e.g., PY14⁺, in the interlayer⁵⁶.”

Figure 5d. A GITT curve for the (de)intercalation of MgCl⁺ into *ex*TiS₂ electrode.

5. *Was the diffusion coefficient of Figure 5(f) measured during discharge or charge? This should be clarified in the text. The reduction in the diffusion coefficient with increasing concentration could be due to a divacancy diffusion mechanism as is well known to occur in Li-intercalation compounds. A relevant reference in this context is: Van der Ven, A.; Bhattacharya, J.; Belak, A. A. Acc. Chem. Res., 46, 1216–1225, 2013*

Our response: We thank the Reviewer for bringing to our attention an important reference in explaining the observed concentration-dependent diffusivity. The diffusion coefficient shown

in the original Figure 5f was measured during discharge (intercalation). Overall, the diffusion coefficient decreases with increasing MgCl^+ concentration. As the reviewer pointed out, this can be due to a divacancy diffusion mechanism as is well known to occur in Li-intercalation compounds¹³. We have included the above discussion and reference in the revised manuscript.

In revision, we have specified that the diffusivity is calculated based on discharge (page 10): “The diffusivity calculated during discharge is initially high at the level of $3 \times 10^{-10} \text{ cm}^2 \text{ s}^{-1}$...” The possible divacancy mechanism is discussed in the same paragraph: “The decrease in the diffusivity with increasing MgCl^+ concentration can be due to a divacancy diffusion mechanism as is well known to occur in Li^+ intercalation compounds⁵⁷.”

References

1. Zhang, Z. *et al.* High energy density hybrid $\text{Mg}^{2+}/\text{Li}^+$ battery with superior ultra-low temperature performance. *J. Mater. Chem. A* **4**, 2277-2285 (2016).
2. Li, Y. *et al.* A high-voltage rechargeable magnesium-sodium hybrid battery. *Nano Energy* **34**, 188-194 (2017).
3. Shterenberg, I. *et al.* Evaluation of $(\text{CF}_3\text{SO}_2)_2\text{N}^-$ (TFSI) Based Electrolyte Solutions for Mg Batteries. *J. Electrochem. Soc.* **162**, A7118-A7128 (2015).
4. Connell, J. G. *et al.* Tuning the Reversibility of Mg Anodes via Controlled Surface Passivation by $\text{H}_2\text{O}/\text{Cl}^-$ in Organic Electrolytes. *Chem. Mater.* **28**, 8268-8277 (2016).
5. Bitenc, J., Firm, M., Randon Vitanova, A. & Dominko, R. Effect of Cl^- and TFSI^- anions on dual electrolyte systems in a hybrid $\text{Mg}/\text{Li}_4\text{Ti}_5\text{O}_{12}$ battery. *Electrochem. Commun.* **76**, 29-33 (2017).
6. Ha, J. H., Cho, J.-H., Kim, J. H., Cho, B. W. & Oh, S. H. 1-Butyl-1-methylpyrrolidinium chloride as an effective corrosion inhibitor for stainless steel current collectors in magnesium chloride complex electrolytes. *J. Power Sources* **355**, 90-97 (2017).
7. Tutusaus, O. *et al.* An efficient halogen-free electrolyte for use in rechargeable magnesium batteries. *Angew. Chem. Int. Ed.* **54**, 7900-7904 (2015).
8. Montanino, M. *et al.* Physical and electrochemical properties of binary ionic liquid mixtures: $(1-x)$ $\text{PYR}_{14}\text{TFSI}-(x)$ $\text{PYR}_{14}\text{IM}_{14}$. *Electrochim. Acta* **60**, 163-169 (2012).
9. Liao, C. *et al.* The unexpected discovery of the $\text{Mg}(\text{HMDS})_2/\text{MgCl}_2$ complex as a magnesium electrolyte for rechargeable magnesium batteries. *J. Mater. Chem. A* **3**, 6082-6087 (2015).
10. Wan, L. F., Perdue, B. R., Apblett, C. A. & Prendergast, D. Mg Desolvation and Intercalation Mechanism at the Mo_6S_8 Chevrel Phase Surface. *Chem. Mater.* **27**, 5932-5940 (2015).
11. Wu, Z. Y. *et al.* Sulfur *K*-edge X-ray-absorption study of the charge transfer upon lithium intercalation into titanium disulfide. *Phys. Rev. Lett.* **77**, 2101-2104 (1996).
12. Yu, H. C. *et al.* Designing the next generation high capacity battery electrodes. *Energy Environ. Sci.* **7**, 1760-1768 (2014).
13. Van der Ven, A., Bhattacharya, J. & Belak, A. A. Understanding Li Diffusion in Li-Intercalation Compounds. *Acc. Chem. Res.* **46**, 1216-1225 (2013).

Reviewers' Comments:

Reviewer #1:

Remarks to the Author:

The authors have addressed all the points I brought up to them. I am satisfied with their responses and have no further concerns.

Reviewer #2:

Remarks to the Author:

This is an original piece of work which already reviewed.

The authors addressed very well the comments of all the 4 reviewers.

I recommend publication of the revised paper as is.

Reviewer #3:

Remarks to the Author:

The authors addressed all the questions and clarified all ambiguities that I raised in the original version of the paper. I would recommend the paper to be published, however, in the new version of the paper in the response to the Reviewer #2, the authors added additional material for correcting the capacity of activated electrodes. They find 1.7 MgCl⁺ intercalated per each TiS₂ at 60°C, which is a huge result if measurements were done properly. This would suggest that Ti is changing its oxidation state from Ti(IV) to Ti(II)! This was not observed before. However, judging from the method the authors used for obtaining the stoichiometry, this could also be a consequence of significant adsorption of Mg on the surface of the electrodes. It would be important that the authors describe in detail the process and the calculations on which they derived this large capacity. I would also strongly suggest that the authors obtain directly the oxidation state of Ti after cycling at 60°C as that would eliminate all possible sources of errors and confirm this huge result.

Response to Reviewers' Comments

We thank all four Reviewers for recommending publication of the revised manuscript. We feel we have addressed the comments from Reviewer #3 (see below) and hope this final revision of the manuscript could be suited for publication.

Reviewer #3

- 1. The authors addressed all the questions and clarified all ambiguities that I raised in the original version of the paper. I would recommend the paper to published, however, in the new version of the paper in the response to the Reviewer #2, the authors added additional material for correcting the capacity of activated electrodes. They find 1.7 MgCl⁺ intercalated per each TiS₂ at 60°C, which is a huge result if measurements were done properly. This would suggest that Ti is changing its oxidation state from Ti(IV) to Ti(ii)! This was not observed before. However, judging from the method the authors used for obtaining the stoichiometry, this could also be a consequence of significant adsorption of Mg on the surface of the electrodes. It would be important that the authors describe in detail the process and the calculations on which they derived this large capacity. I would also strongly suggest that the authors obtain directly the oxidation state of Ti after cycling at 60°C as that would eliminate all possible sources of errors and confirm this huge result.*

Our response: We want to thank the reviewer for finding our results important and recommend for publication. Here we clarify the detailed process and the calculations on which we derived the large capacity of 1.7 MgCl⁺ per unit TiS₂. Before the electrochemical cycling, all the cells were activated at 25 °C by discharging the TiS₂/Mg cells to 0.0 V vs Mg/Mg²⁺ at 5 mA g⁻¹ for 100 hours and subsequent cycling the cells within 0.0–2.0 V vs Mg/Mg²⁺ at 24 mA g⁻¹ for 10 cycles (as shown in Fig. 3a). Then the activated *ex*TiS₂/Mg cells were cycled within 0.0–2.0 V vs Mg/Mg²⁺ at varied temperature from –45 to 60 °C. The capacity was specified based on the mass of TiS₂ in the electrode; the specific capacity of 400 mAh g⁻¹ translated to 1.7 MgCl⁺ per TiS₂ by comparing with the theoretical value of 239.3 mAh g⁻¹ for the intercalation of 1.0 MgCl⁺ into TiS₂. EDS also confirms this result. To test if the large capacity comes from the genuine intercalation rather than the adsorption of ions on the surface of the electrodes, we conducted X-ray absorption spectroscopy (XAS) at Ti *K*-edge and S *K*-edge (Fig. R1 and R2). The preliminary data evidences the shift of Ti *K*-edge to lower energy upon discharging the cell at 60 °C, which suggest the reduction of Ti by the intercalation of 1.7 MgCl⁺ per TiS₂ (Fig. R1). It is noteworthy to mention the case of Mg intercalation into Chevrel phase molybdenum sulfide (Mo₆S₈), which reports almost no changes in Mo *K*-edge while S *K*-edge exhibits significant changes upon intercalation of 2.0 Mg²⁺ per Mo₆S₈ (Ref. 1). In line with the literature, clearer changes were observed with S *K*-edge spectra on (MgCl)_{*x*}*ex*TiS₂, which evidences the continuous decrease in the relative intensity of *t*_{2g} level as the lowest level of Ti 3*d* orbitals are partially filled upon the reduction of Ti atom by intercalation of ~1.6 MgCl⁺ per TiS₂ (Fig. R2). Although these are preliminary *ex-situ* results obtained in a limited time span of two weeks, we believe that they provide sufficient evidence to address the reviewer's concern. However, we found that it is premature to include this data in this manuscript because we plan to conduct a more detailed study using

in operando Ti *K*-edge and S *K*-edge XAS as we will have XAS beam time at APS in September 2017.

In revision, we have included the detailed description of the experimental and calculation procedures in the Methods section at page 16: “Before the electrochemical cycling, all the cells were activated at 25 °C by discharging the TiS₂/Mg cells to 0.0 V vs Mg/Mg²⁺ at 5 mA g⁻¹ for 100 hours and subsequent cycling the cells within 0.0–2.0 V vs Mg/Mg²⁺ at 24 mA g⁻¹ for 10 cycles (as shown in Fig. 3a). Then the activated *ex*TiS₂/Mg cells were cycled within 0.0–2.0 V vs Mg/Mg²⁺ at varied temperature from –45 to 60 °C. The capacity was calculated based on the mass of TiS₂ in the electrode; the specific capacity of 400 mAh g⁻¹ was translated to 1.7 MgCl⁺ per TiS₂ by comparing with the theoretical value of 239.3 mAh g⁻¹ for the intercalation of 1.0 MgCl⁺ into TiS₂.”

Figure R1. XAS spectra at Ti *K*-edge for (MgCl)_{1.7}*ex*TiS₂ discharged at 60 °C compared with pristine TiS₂.

Figure R2. XAS spectra at S K -edge for $(\text{MgCl})_x\text{exTiS}_2$ for varied $x = 0.0, 0.5, 1.0,$ and 1.6 .

References

1. Wan, L. F. *et al.* Revealing electronic structure changes in Chevrel phase cathodes upon Mg insertion using X-ray absorption spectroscopy. *Phys. Chem. Chem. Phys.* **18**, 17326-17329 (2016).